

# Environmental changes, climate and anthropogenic impact in southern-eastern Tunisia during the last 8 kyr

Sahbi Jaouadi[1], Vincent Lebreton[1], Viviane Bout-Roumazeilles[2], Giuseppe Siani[3], Rached Lakhdar[4], Ridha Boussoffara[5], Laurent Dezileau[6], Nejib Kallel[7], Beya Mannai-Tayech[8], Nathalie Combourieu-Nebout[1]

[1]UMR 7194 CNRS, Histoire naturelle de l'Homme Préhistorique, Département de Préhistoire, Muséum national d'Histoire naturelle, Paris, France
[2]Laboratoire d'Océanologie et de Géosciences LOG, UMR8187, CNRS-Université Lille-Université Cote d'Opale, 59655 Villeneuve d'Ascq, France
[3]Laboratoire des Interactions et Dynamique des Environnements de Surface (IDES), UMR8148, CNRS-Université de Paris-Sud, Bat 504, 91405 Orsay Cedex, France
[4]Faculté des Sciences de Bizerte, Université de Carthage, 7021 Zarzouna, Bizerte, Tunisia
[5]Institut National du Patrimoine. 4 Place du Château, 1008 Tunis, Tunisia
[6]UMR 5243 CNRS, Géosciences Montpellier, Université de Montpellier, Montpellier, France
[7]Université de Sfax, Faculté des Sciences, Laboratoire GEOGLOB, BP 802, 3038 Sfax, Tunisia
[8]Université de Tunis El Manar, Faculté des Sciences de Tunis, 2092, Tunis, Tunisia

*Correspondence to*: Sahbi Jaouadi (jaouadisahbi@yahoo.com)

**Abstract.** Pollen and clay mineralogical analyses of a Holocene sequence from Sebkha Boujmel (southern Tunisia) traces the climatic and environmental dynamics in the lower arid bioclimatic zone over the last 8000 years. During the Mid- to Late Holocene transition, between 8 and 3 ka, a succession of five wet/dry oscillations is recorded. An intense arid event occurs between 5.7 and 4.6 ka. This episode marks the onset of a long-term aridification trend with a progressive retreat of Mediterranean woody xerophytic vegetation and of grass steppes. It ends with the establishment of pre-desert ecosystems around 3 ka. The millennial-scale climate change recorded in the data from Sebkha Boujmel is consistent with records from the south and east Mediterranean, as well as with climatic records from the desert region for the end of the African Humid Period (AHP). Eight centennial climatic events are recorded at Sebkha Boujmel and these are contemporary with those recorded in the Mediterranean and in the Sahara. They indicate a clear coupling between the southern Mediterranean and the Sahara before 3 ka. The event at 4.2 ka is not evidenced and the link between events recorded in Sebkha Boujmel and the North Atlantic Cooling events is clearer from 3 ka onwards. These variations indicate the importance of climatic determinism in the structuring of landscapes, with the establishment of the arid climatic conditions of the Late Holocene. It is only from 3 ka onwards that the dynamic of plant associations is modified by both human activity and climatic variability. The climatic episodes identified during the historic period indicate strong regionalisation related to the differential impact of the North Atlantic Oscillation (NAO) and the Mediterranean Oscillation (MO) on the Mediterranean basin. The local human impact on regional ecosystems is recorded in the form of episodes of intensification of pastoral and/or agricultural activities. The development of olive production and of several taxa associated with agriculture attest to increasing sedentism among human populations during Classical Antiquity. The significant increase in *Artemisia* (wormwood) between 1.1 and 0.8 ka





(850 – 1150 AD) is linked to intensive pastoral activity, associated with heightened interannual and/or seasonal climatic instability. A complete re-shaping of the landscape is recorded during the 20th century. The remarkable expansion of the olive tree, and the deterioration of regional ecosystems with the spread of desert species, is linked to recent local socio-economic changes in Tunisia.

**1 Introduction**

Subjected to climatic influences from both polar high latitudes and subtropical low latitudes, the Mediterranean Basin is considered to be one of the regions most exposed to climatic change (IPCC, 2014). Its ecosystems are particularly vulnerable to hydrological changes as well as to strong demographic pressure, particularly in coastal areas (Lionello et al., 2006; IPCC, 2014). In this context, the desert margins and arid ecosystems of North Africa constitute climatic and biogeographical

transition zones that are among the most sensitive to variations in climatic parameters and human activities (Smykatz-Kloss and Felix-Henningsen, 2004). Among these regions, south-eastern Tunisia is currently crippled by arid conditions and increasing human pressure, leading to a marked degradation of its natural environment and increased desertification (Floret and Pontanier, 1982; Genin et al., 2006). Moreover, prospective studies foresee a prolonging of drought periods and intensified desertification due to the effects of global warming (e.g. Gao and Giorgi, 2008; Giorgi and Lionello, 2008; Nasr

et al., 2008; Giannakopoulos et al., 2009).

In the northern Mediterranean, numerous palaeoecological records reveal the climate dynamics for the Middle to Late Holocene (e.g. Roberts et al., 2011; Sadori et al., 2011; Magny et al., 2013; Azuara et al., 2015). For the Sahara, the scale of Holocene climate change, with the establishment of a hyper-arid desert, has been estimated using multi-proxy analyses (e.g. deMenocal et al., 2000; Swezey, 2001; Gasse and Roberts, 2004; Hoelzmann et al., 2004; Kröpelin et al., 2008; Lézine et al.,

2011; McGee et al., 2013; Tierney and deMenocal, 2013; Cremaschi et al., 2014). However, palaeoecological and palaeoclimatic records from the southern Mediterranean are still needed in order to complete the pattern of environmental and climatic changes along a latitudinal transect in the Mediterranean region. In fact, during the Early and Middle Holocene, the climatic and hydrological contrast between the northern and southern parts of the Mediterranean basin is already apparent (Magny et al., 2012, 2013; Peyron et al., 2013). Regional-scale climate modelling simulations spanning the

Holocene also reveal a climatic contrast between the east and west shores of the Mediterranean, along a line passing between Italy and Tunisia (Brayshaw et al., 2011). Today, there is still a dearth of available data for the southern shore of the Mediterranean, and in particular for the arid regions of North Africa, thus providing only a fragmentary view of Holocene climate changes. Indeed, the scarcity of suitable deposits for pollen analyses in the arid regions has limited the acquisition of palaeoecological data (Horowitz, 1992; Carrión, 2002b). The lack of data for semi-arid and arid bioclimatic areas has been

highlighted in Holocene climatic syntheses in the Mediterranean (Tzedakis, 2007; Roberts et al., 2011), leading to geographical gaps in the outputs from climate modelling simulations (Brayshaw et al., 2011; Peyron et al., 2011).



In the pre-desert steppes of the Maghreb, data regarding lake-level variations (Gasse, 2000; Callot and Fontugne, 2008), geomorphological features (e.g. Ballais, 1991; Ballais and Ouezdou, 1991; Boujelben, 2015), sand dune dynamics (Swezey et al., 1999; Swezey, 2001), and pollen sequences (Brun, 1983, 1992; Damblon and Vanden Berghen, 1993; Salzmann and Schulz, 1995), are characterised by low temporal resolutions and are geographically and chronologically dispersed. These

data indicate a more humid climate during the Early and Middle Holocene followed by an increase in aridity over the course of the Late Holocene.

Analysis of past environmental dynamics in pre-Saharan Tunisia is crucial and should allow the characterisation of: (1) the biogeographical history of the present landscape, (2) the resilience of vegetation in response to aridity and human disturbance and (3) the processes degradation and desertification of desert margins (Schaaf, 2008). Recent work undertaken

on the continental Sebkha Mhabeul in southern Tunisia (Schulz et al., 2002; Marquer et al., 2008) and the Halk el Menjel sebkha-lagoon in central Tunisia (Lebreton and Jaouadi, 2013; Lebreton et al., 2015) has already revealed the huge potential of sebkhas for pollen analysis. Sebkha Boujmel, on the southern coast of Tunisia, has yielded the first continuous geochemical and pollen record for the last 8000 years. The sequence reveals the vegetation history in the transition zone between the Mediterranean and the Sahara. In this study, vegetation dynamics and climatic changes are integrated in a

transect of the central Mediterranean in order to identify their rhythmicity and to correlate them with regional and global climate changes. The human impact is considered within the cultural framework of Tunisian archaeology and history.

## 2 Geographical settings, climate and current vegetation

Today, south-eastern Tunisia is dominated by an arid Mediterranean climate having a long dry season with occasional and highly irregular precipitation. Two hydro-climatic gradients are evident from north to south and from east to west, under the

combined influence of proximity to the Mediterranean coast and upland topography, which captures humid air masses (Berndtsson, 1989). Three climatic and phyto-geographical territories have been identified for south-eastern Tunisia: the Matmata mountain range, the desert, and the Jeffara pre-desert coastal plain (Le Houérou, 1959, 1995; Floret and Pontanier, 1982; Genin et al., 2006) (Fig. 1b).

In the Matmata Mountains zone, to the west, increased humidity on relief up to 700m produces an altitudinal distribution of

vegetation (Fig. 1b). The northern summits of the Matmata Mountains, endowed with a more humid climate (with average precipitation values of between 200 and 300 mm per annum) are part of the upper arid bioclimatic zone. On these summits, a low shrubland of *Junepirus phoenicea* and *Rosmarinus officinalis* develops locally. This vegetation gradually gives way to grassy and semi-desert steppe to the east and south due to increased aridity as one descends into the foothills (Fig. 1b).

A gradual transition towards the desert area is observed as soon as humidity, due to proximity to the Mediterranean and the

uplands, recedes. A desert bioclimate, with annual precipitation below 100mm, borders on the Matmata mountainous zone and the Jeffara coastal plain with the sand dunes of the Great Eastern Erg to the west and the Sahara to the south (Fig. 1b). Vegetation is sparse and adapted to the arid conditions with psammophyte shrubs (*Calligonum* sp., *Ephedra alata* subsp.



*alenda* and *Retama raetam*) and desert herbaceous plants such as Amaranthaceae (*Cornulaca monacantha, Traganum nudatum*), Boraginaceae (*Echium* sp., *Moltkiopsis ciliata*), Zygophyllaceae (*Fagonia* sp., *Nitraria retusa*), Brassicaceae (*Henophyton deserti*) and Euphorbiaceae (*Euphorbia guyoniana*).

Further east, the Jeffara coastal plain, which forms an extensive pre-desert area delimited at the west by the Matmata Mountains, slopes gradually down to the coast (Fig. 1b). The coastal area features a large number of lagoons and coastal sebkhas (Fig. 1c). The Jeffara Plain forms part of the Mediterranean lower arid bioclimatic zone, with mild winters and average annual precipitation of between 100 and 200 mm. It is occupied by semi-desert steppe, featuring *Rhanterium suaveolens*, *Artemesia sp.*, *Haloxylon scoparium, Gymnocarpos decander,* and degraded grassland steppe with *Stipa tenacissima* on the foothills and plains with a calcareous crust. A halophytic crassulescent steppe, characterised by Amaranthaceae (*Halocnemum*, *Salicornia*, *Salsola* and *Suaeda*), develops in the saline soils around the sebkhas. On the Jeffara Plain, the anthropogenic impact is significant and includes ancestral pastoralism as well as recent conversion to agriculture. The Jeffara was originally used as rangeland for nomadic tribes and their flocks. The recent anthropogenic impact takes the form of vast olive groves (Fig. 1b). In this area, the olive is outside its natural bioclimatic zone and its cultivation is made possible through dry-farming and traditional systems for controlling surface run-off in the mountains and foothills (Nasri et al., 2004).

The Sekhba Boujmel (33°16'N, 11°05'E, 2 m. asl) is situated on the coast of the Jeffara Plain and forms part of the paralic complex of the Bibane Lagoon (Fig. 1c). This supratidal sekhba is irregularly flooded by rainwater and high tides via the Bibane Lagoon. The sekhba Boujmel and Bibane Lagoon correspond to valleys which were incised by the Wadi Fessi during the Upper Pleistocene (Medhioub and Perthuisot, 1981). The Sebkha Boujmel corresponds to the present-day and Holocene delta of the Wadi Fessi, which rises inland at Tatouine then drains the mountainous region and the coastal plain (Keer, 1976; Medhioub and Perthuisot, 1981) (Fig. 1b). The Holocene sedimentary filling of the Sebkha Boujmel is a typical lagoonal sequence with biodetrital deposits at the base, followed by bioclastic and oolithic carbonate sands and, finally, by detrital facies which are principally of eolian origin (Fig. 2) (Lakhdar et al., 2006).

## 3 Material and methods

### 3.1 Core material and samples

The BJM2 core (33°18'30.96"N, 11° 5'0.68"E, 160 cm depth) was extracted from the north-west of the sebkha in order to carry out pollen and mineralogical analysis (Fig. 1c). The observed effect of core shortening (12.5 %), which commonly occurs in shallow unconsolidated coastal sediments, has been removed using field measurements following the method described in Morton and White (1997). 71 samples were selected for palynological and clay mineralogical analyses.





### 3.2 Chronology

The chronology of the BJM2 core is based on eleven AMS $^{14}$C dates carried out on the organic fraction of the sediment (Table 1). Study of the organic matter shows that it is of mixed origin, composed of marine planktonic/algal material and continental woody material (Lakhdar, 2009). Input of marine origin is also confirmed by the extensive presence along the core of shallow-dwelling benthic foraminifera of the genus *Ammonia* as well as Ostracoda and Mollusks. The $^{14}$C dates were corrected for the reservoir effect, calculated at 400 yrs for the Mediterranean (Siani et al., 2001), and then calibrated using IntCal13 within the Calib 7.1 programme (Reimer et al., 2013) (Table. 1). The date at the top of the sequence, considered to be predominantly under continental influence, was not corrected for the reservoir effect.

The age-depth model (Fig. 2) was elaborated using the Clam software package (Blaauw, 2010; Blaauw and Heegaard, 2012). Models with age reversals were rejected and best goodness-of-fit was obtained using 3$^{rd}$ degree polynomial regression. The absence of sedimentary hiatus (Lakhdar, 2009), the number of AMS $^{14}$C dates, and the coherence of the age-depth model, limit the influence of erroneous dates or dates impacted upon by a hard water effect.

The BJM2 core records 8000 years of infill history of the Sekhba Boujmel. The sampling resolution varies as a function of the sedimentation rate. It is around 210 yrs/sample for the period between 8 and 3 ka and with an average of 60 yrs/sample for the rest of the sequence.

### 3.3 Pollen analysis

The pollen extraction follows the standard protocol (Traverse, 2007): 5g of sediment are treated by successive chemical attacks (HCl 18 %, HF 70 % and HCl 10 %), followed by ultrasonic filtration at 5µm. A tablet, calibrated for *Lycopodium* spores, is added to each sample at the start of the treatment in order to estimate the pollen concentration (Stockmarr, 1971). Taxonomic determinations are carried out using a Zeiss microscope under x1000 magnification. Identifications follow published pollen atlases (Reille, 1992, Ayyad and Moore, 1995; Beug, 2004) and pollen reference collections of the Prehistory Department of the Muséum National d'Histoire Naturelle and the Institut des Sciences de l'Evolution of Montpellier (ISEM). An average of 300 pollen grains are counted for each sample, and then completed by research of entomogamous taxa that are common in arid regions and under-represented in pollen assemblages (Horowitz, 1992; Carrión, 2002b).

The pollen/vegetation relationship and the climatic affinities of the pollen flora are established on the basis of specialised flora (Le Houérou, 1959, 1969, 1980, 1995; Pottier-Alapetite, 1979, 1981; Floret and Pontanier, 1982; Chaieb and Boukhris, 1998; Ozenda, 2004), completed by surface pollen spectra which are representative of the different regional environments (Jaouadi et al., accepted). The Amaranthaceae are a significant floristic component of the desert margins of southern Tunisia. Several botanic genera occupy either desert habitats or saline hollows along the coast. In order to separate the regional xerophytic from the local halophytic signal, the pollen morphology of the principal genera of desert Amaranthaceae in southern Tunisia (*Anabasis*, *Cornulaca*, *Haloxylon*, *Traganum*) has been studied with reference to the large ISEM reference




collection and to previously published works (Nowicke, 1975; Nowicke and Skvarla, 1979; Salzmann and Schulz, 1995). Two pollen types have thus been distinguished: Amaranthaceae-type, which essentially groups together the halophytic taxa, and *Cornulaca-Traganum*-type which principally encompasses the xerophytic genera.

The pollen data are presented in the form of a simplified diagram of the most significant taxa grouped together in three

ecological units, which are characteristic of the regional vegetation: Mediterranean xerophytes, steppic herbaceous plants, and desert herbaceous plants (Fig. 3). The percentages are calculated with respect to a basic sum that only includes these three groups. The percentages of aquatic-, Amaranthaceae-type-, long-range transported-, cultivated-, nitrophilous- and introduced taxa are calculated on the basis of a total sum of pollen grains identified within each pollen spectrum (Berglund and Ralska-Jasiewiczowa, 1986). *Olea* was not included in the basic sum and is included in the cultivated taxa as soon as its

record coincides with that of the latter (Fig. 3 and Fig. 5). The Wet/Dry ratio (R/D) developed by Hooghiemstra (1996) is applied to the Sekhba Boujmel pollen data (Fig. 4n) in order to trace the changes between the xerophytic herbaceous vegetation of the steppe and desert (Dry amount = Asteraceae excluding *Artemisia* + Amaranthaceae *Cornulaca-Traganum* type) and the steppic grasses that develop in more humid conditions (Wet amount = Poaceae + Cyperaceae) (Hooghiemstra, 1996; Mercuri, 2008; Giraudi et al., 2013; Cremaschi et al., 2014).

The coherence and correlation between the ecological groups, the taxa used for the W/D ratio, the pollen data and the clay mineralogical data, are provided by a Correlogram reordered based on the first axis of the Principal Component Analysis (Friendly, 2002). The processing of the pollen data is carried out using the "Rioja" package of the "R" software (R Core Team, 2013; Juggins, 2015). The Local Pollen Zones (LPZ) are determined by visual observation of the pollen diagram and are validated by applying constrained hierarchical clustering by CONISS (Fig. 3) (Grimm, 1987). Long term changes in the

vegetation are represented by applying a cubic smoothing spline method to the data (Fig. 4).

### 3.4 Clay mineralogy

A total of 68 samples were subjected to qualitative and semi-quantitative analysis of the clay fraction (< 2 μm) using x-ray diffraction. This procedure was carried out using a Bruker D4 Endeavor diffractometer coupled with a Lynxeye rapid detector fitted with a copper anticathode. The samples were prepared using a standard protocol (Bout-Roumazeilles et al.,

1999). The samples are broken down in distilled water, decarbonated using HCl N/5, and then deflocculated by repeated rinsing with distilled water. The suspensions obtained are placed in pill-boxes and the micro-aggregates eliminated by a microhomogeniser. The granulometric fraction below 2 μm is separated by collecting the upper part of the suspension after 1hr 15mins of decantation, and is then centrifuged. Diffractometric analysis is carried out on three preparations: untreated, glycolated, and heated. Clay mineral identification is carried out on the three preparations (Brindley and Brown, 1980). The

semi-quantitative analysis is based on the integration of the signals for the main peaks of the clay minerals using the MacDiff software programme (Petschick, 2001).



## 4 Results

### 4.1 Pollen data

The pollen record for the Sebkha Boujmel features good pollen grain preservation, with an average pollen concentration of ca. 35000 grains per $cm^{-3}$, and wide taxonomic diversity with 114 pollen taxa being recorded.

For the Holocene, the Sebkha Boujmel, which was fed by the Wadi Fessi, has revealed fluvial and eolian pollen inputs from local and regional vegetation (Fig. 1b). *Pinus* is never overrepresented, indicating a moderate maritime influence and pollen inputs from a catchment area that is limited to the Jeffara Plain, the Matmata highlands and the desert margins. However, a few non-native pollen grains from the temperate regions of North Africa and continental Europe (*Alnus*, *Betula*, *Corylus*, *Carpinus*/*Ostrya*, *Cedrus* and *Myrica*) have been identified throughout the sequence. These taxa, which are also recorded in

other desert regions, demonstrate long-distance transportation of pollen by atmospheric circulation from the Mediterranean to the desert regions (Cour and Duzer, 1980; Schulz, 1984). In fact, the diversity in the pollen assemblages from the Sebkha Boujmel reflects the local steppic zones, the mountainous hinterland and the desert region.

Seven Local Pollen Zones (LPZ) retrace the vegetation history in response to climate changes and the impact of human activity over the Middle and Late Holocene (Table 2, Fig. 3). The sequence from Sebkha Boujmel documents the landscape

dynamic of southern Tunisia, from the Mediterranean shrub-lands to the pre-desert steppes on the uplands, as well as the spread of pre-desert steppes on the plains. These vegetation changes reflect the establishment of increasingly arid conditions since the Middle Holocene, on top of which is added increasing human pressure over recent centuries (Table 2, Fig. 3 and 5).

### 4.2 Clay composition and origins

Mineralogical analyses indicate that the clay fraction of the sediments is composed, on average, of 40 % kaolinite, associated

with 25% palygorskite and 25% illite. Chlorite and smectite are less abundant and represent 7% and 2% of the clay assemblage respectively. All clay minerals could have been transported to the Sebkha Boujmel via eolian processes or via the Wadi Fessi fluvial system. Both processes are rather irregular, and are seasonnaly modulated by the precipitation regime. Illite mainly results from the process of physical erosion. It is often abundant in wind-blown particles and indicates a Saharan origin when it is associated with palygorskite (Foucault and Mélières, 2000; Goudie and Middleton, 2001).

Palygorskite is a mineral that is typical of arid and semi-arid Mediterranean environments, which are characterised by alternating dry and humid periods. Palygorskite is a fibrous clay mineral abundant in wind-blown deposits and rare in river sediments since the elongated palygorskite particles are easily broken during fluviatile transportation. It is abundant in northern Saharan desert dusts and in Tunisian dunes and loess, particularly Matmata loess (e.g. Coudé-Gaussen and Rognon, 1988; Bout-Roumazeilles et al., 2007). In North Africa, kaolinite is more abundant in the central and southern Sahara, in

Sahelian zones, where it is produced through intense hydrolysing processes, and in the eastern Sahara than in the western Sahara (e.g. Caquineau et al., 1998). As a result, an increase in the proportion of kaolinite at the expense of illite in the eolian fraction suggests a dominant Sahelian origin for the clay particles (O'Hara et al., 2006; Hamann et al., 2009). The illite to





kaolinite ratio (I/K) is, therefore, used to distinguish the respective contributions from Saharan and Sahelian sources in eolian deposits (Fig. 4p) (e.g. Caquineau et al., 1998; Formenti et al., 2011). Comparison of the Sebkha Boujmel samples with those from North African sands and sediments in terms of the I/K ratio and the percentage of palygorskite, indicates that the sediments are a mix of inputs of Tunisian loess (I/K= 1.5 and 45% palygorskite) and inputs from the Saharan zone (I/K = 0.4-2; 5-25% palygorskite). The rather high I/K ratio observed in the Sebkha Boujmel sediments, ranging between 0.5 and 0.9, rules out any major contribution from the Sahelian region (I/K = 0.1), but is consistent with mixed contributions from the south/central Sahara (I/K= 0.4) and the north/west Sahara (I/K= 0.5-0.7, up to 2) (e.g. Caquineau et al., 1998).

## 5 Discussion

### 5.1 Biogeography and Holocene palaeoenvironment in southern Tunisia

Modern pollen rain from the vicinity of the sebkha and the mountainous hinterland tie in well with the regional biogeogeography. In these arid areas, the modern pollen spectra display a very low presence of Mediterranean taxa that require more humid conditions (Jaouadi et al., accepted). Today, only degraded relics of Mediterranean vegetation still survive on the wetter northern peaks of the Matmata Mountains (Le Houérou, 1959, 1969, 1995; Pottier-Alapetite, 1979, 1981; Chaieb and Boukhris, 1998). However, Mediterranean vegetation, and particularly *Pistacia*, appears to have been well developed at the beginning of the Middle Holocene (Fig. 3). The uplands to the west of the Sekhba Boujmel probably permitted the development of Mediterranean vegetation in southern Tunisia, alongside the Matmata Mountains at least as far as the latitude of Tataouine (Fig.1b), during wetter climatic episodes. However, the Mediterranean vegetation did not extend as far as the mountainous areas of Libya further south. In the Libyan part of the Jeffara, the woody species that developed at Jbel Gharbi between 9.4 and 5 ka constitute a desert-adapted shrubland formation (*Capparis*, *Ficus*, *Salvadora persica* and *Tamarix*) in which Mediterranean taxa remain poorly represented (Giraudi et al., 2013). Here the regional geographical features play an important role in limiting the southward progression of Mediterranean elements to the Tunisian Jeffara Plain because of their ecological requirements. Thus, the structure of the Mediterranean flora recorded at Sebkha Boujmel exhibits a significant xerophytic component, even during wetter periods, with *Pinus*, *Juniperus* and, in particular, *Rhus tripartita*, a Saharo-Mediterranean shrub. In addition, significant latitudinal spreads of Mediterranean vegetation may have occurred between central and southern Tunisia during favourable climatic episodes in the Holocene. The Holocene development of thermophilic bushland, with wild olive (*Olea europaea* spp. *europaea* var. *sylvestris*) and pistachio (*Pistacia lentiscus*), is thus well documented in the Gulf of Gabes (Brun, 1992). The maximum extension of the *Oleo-lentiscetum* on the coast of the Gulf of Hammamet occurs at the 400 mm isohyet, which marks the limit of the semi-arid bioclimatic stage (Lebreton et al., 2015). At Sebkha Boujmel, *Olea* is recorded with low rates, reflecting a presence limited to the wettest biotopes. The *Pistacia* pollen taxon might correspond principally to *P. atlantica*. *P. atlantica* is more xerophytic than *P. lentiscus* and *O. europaea spp. europaea var. sylvestris*, with a wider distribution stretching from the sub-humid area to the Saharan area (Le Houérou, 1969). Moreover, its presence fits well with the structure of the pollen flora recorded at the Sebkha Boujmel.



Taking these biogeographical elements into account, over the course of the Middle Holocene, the base of the sequence indicates an open landscape with grass steppe in the coastal lowlands and woody steppe, featuring *Pistacia atlantica*, *Juniperus phoenicea* and *Rhus tripartita,* on the foothills and along the waterways. Towards the northwest, the wetter highlands of the Matmata Mountains (Fig.1b) would have been occupied by *Pistacia lentiscus*, *Olea europaea spp. europaea var. sylvestris* and *Cistus*. It is also possible that *Quercus ilex* occurred towards the north on the more favourable mountain slopes. The regional landscape of the plain would have remained relatively open with herbaceous steppes among which grass steppes were predominant. The arid phases are attested to in the development of xerophytic Amaranthaceae while *Artemisia*, although present, is never very abundant during the Middle Holocene.

**5.2 Mid- to Late Holocene climate changes and landscape evolution**

**5.2.1 Aridity trend and the Mid- to Late Holocene transition**

Between 5.7 and 4.6 ka, pollen data reveal a major shift towards the progressive establishment of steppic and pre-desert ecosystems, within which Mediterranean trees and shrubs play no role (Fig. 4). These environmental changes reflect the considerable impact of the climate during this period. A significant drop in humidity is recorded between 5.7 and 4.6 ka, with two high-amplitude arid events (see Fig. 4, BJ6: 5.75 – 5.5 ka and BJ5: 4.9 – 4.6 ka). The abrupt decrease in pollen grains from aquatic plants reflects reduced precipitation associated with a reduction in permanent and/or seasonal water bodies (Fig. 4). Such drastic change is accompanied by a drop in the W/D (wet/dry) and I/K (illite/kaolinite) ratios which attests to greater aridity and enhanced mobilization of fine particles over long distances, probably from the central Sahara (Fig. 4). These arid events are of major significance because they occur during a period that was globally more humid. Subsequently, from about 4.5 ka onwards, pollen data and clay mineralogy indicate short-term, wet climatic events that are, in fact, phases where there is a slight tendency towards greater humidity during a period which, on the whole, is drier (Fig. 4). These observations could alternately be interpreted as episodic pulses in the Wadi Fessi fluvial system. Following a return to relatively wet conditions around 4.5 to 4 ka, aridity becomes progressively more pronounced from 3.9 ka onwards, with a continuous and irreversible decrease in the percentages of pollen grains from aquatic plants and Mediterranean species as well a drop in the W/D and I/K ratios (Fig. 4). The co-eval decrease in the I/K ratio and palygorskite content (from 35% to 20%) suggests a decreased contribution of Tunisian loess and an enhanced contribution from the central Sahara. Several records, from the Mediterranean and from the Sahara, reveal similarities with those from the Sebkha Boujmel and indicate a significant trend towards aridification between 5.7 and 4.6 ka (Fig. 4). This dynamic is also evidenced at Chott Rharsa where a definitive drop in the water table was accompanied by renewed sand dune formation after 5.6 ka (Swezey et al., 1999; Swezey, 2001). In the Libyan part of the Jeffara Plain, a return of eolian sedimentation occurs between 5.6 and 5.4 ka, followed by the end of the humid period at around 5 ka (Giraudi et al., 2013). On the northern margins of the Algerian Sahara, the definitive drying up of the Holocene palaeolakes, dated to ca. 5.2 ka (Callot and Fontugne, 2008), is followed by the formation of a calcareous crust between ca. 4.5 and 3ka at Hassi el Mejnah (Gasse, 2000, 2002). Further south, the



establishment of hyper-arid climate is evident in the pollen data from Wadi Teshuinat and the Takarkori rock shelters in the Central Sahara between 5.7 and 4.6 ka (Mercuri, 2008; Cremaschi et al., 2014). Further north, in central Tunisia, significant changes in climatic conditions and ecosystems occur simultaneously with: 1) the end of the growth phase of a stalagmite at La Mine Cave at ca. 5.6 ka (Genty et al., 2006); 2) the transition from a permanent to a temporary lake at Sebkha Kalbiyya

5 after 6 ka (Boujelben, 2015); and 3) an arid episode between 5.5 and 5 ka associated with the development of *Pistacia,* based on pollen data from the Halk el Menjel sebkha-lagoon (Lebreton and Jaouadi, 2013). A transition to arid climate is also revealed by isotopic data from Gueldaman Cave in northern Algeria, with a bipartite arid phase identified between 5.7 and 5.2 ka (Ruan et al., 2016). In the south-western Mediterranean, pollen and limnological data from Lake Siles in southern Spain also show a major phase of dessication at around 5.4 – 4.8 ka (Carrión, 2002a). At the same time, between 5.5 and 4.5

10 ka, a major arid phase is recorded in the sediments and marine cores from the Alboran Sea (Combourieu Nebout et al., 2009; Fletcher et al., 2012), the Adriatic Sea (Combourieu-Nebout et al., 2013) and the Siculo-Tunisian Strait (Desprat et al., 2013) (Fig. 4). In the eastern Mediterranean, synthesis of available climatic records also indicates a period of aridification beginning about 5.4 ka and culminating in complete aridity around 4.6 ka (Finné et al., 2011).

All of these contemporaneous events seem to indicate climatic mechanisms at the scale of the southern and eastern

Mediterranean, with significant influence from the African monsoon. In the south, the desert zone is dominated by the seasonal northward shift of the Intertropical Convergence Zone (ITCZ) and of the African monsoon system. Furthermore, the latter is reinforced, from the Early Holocene onwards, by a change in the orbital precession and an increase in summer insolation, which together give rise to the African Humid Period (AHP) (e.g. Kutzbach and Liu, 1997; deMenocal et al., 2000). However, there is still controversy about the date and amplitude of the end of the AHP. Some researchers opt for an

abrupt end (deMenocal et al., 2000; McGee et al., 2013; Tierney and deMenocal, 2013) while others favour a gradual transition towards a hyper-arid climate (Gasse, 2002; Hoelzmann et al., 2004; Kröpelin et al., 2008; Lézine et al., 2011). The significant and rapid drop in humidity recorded in the Sebkha Boujmel sediments between 5.7 and 4.6 ka reflects a dynamic which is similar to the climatic history recorded in the Sahara and the abrupt end of the AHP between 5.3 and 4.5 ka (McGee et al., 2013; Tierney and deMenocal, 2013). Furthermore, such a timing ties in with the Rapid Climate Change (RCC), taking

place between 6 and 5 ka, which marks the end of the humid period in tropical Africa (Mayewski et al., 2004) and greater climatic variability in the western Mediterranean (Fletcher and Zielhofer, 2013). Subsequent changes in the ecosystems and regional biogeography of southern Tunisia, such as those recorded at the Sebkha Boujmel between 4.5 and 3 ka, were progressive and reflect the different reactions of plant communities to the major changes in climatic conditions previously recorded between 5.7 and 4.6 ka. Aquatic plants, for example, which are highly sensitive to changes in humidity, responded

rapidly to these climatic changes, while Mediterranean and herbaceous species reacted more slowly due to their buffering capacity and/or their upland habitat.

However, this temporal similarity does not imply the direct influence of the low-latitude monsoon system and a tropical origin for the moisture recorded on the northern edges of the Sahara over the course of the Holocene. In fact, the influence of the African monsoon never seems to have reached the Mediterranean coast (e.g. Arz et al., 2003; Tzedakis, 2007) and the



stable isotope composition of groundwaters of the Great Eastern Erg suggest winter precipitation of northern origin, which is comparable to present-day winter precipitation resulting from the southward shifting of the mid-latitude westerlies (Gasse, 2002; Guendouz et al., 2003; Edmunds et al., 2004).

At the Sebkha Boujmel, the humid/arid transition, which is contemporary with the end of the AHP, may have resulted from
5 change in the configuration of global climatic drivers, particularly orbital parameters regulating insolation and Mediterranean atmospheric conditions. Changes in atmospheric and ocean circulation are mainly controlled by the Earth's orbital forcing (obliquity and precession) which acts on the climate system by controlling the seasonal and latitudinal distribution of insolation and temperature (e.g. Davis and Brewer, 2009; Brayshaw et al., 2010). Climate simulations indicate that enhanced seasonality of Early Holocene insolation, with minimum precession and maximum obliquity, generates i) an
10 enhancement of the African monsoon, and ii) increased storm track activity and winter precipitation over the Mediterranean (Brayshaw et al., 2011; Kutzbach et al., 2014; Bosmans et al., 2015). According to Kutzbach et al. (2014), the influence of these orbital parameters (summer insolation maxima and winter insolation minima) could have generated a Holocene humid period in the Mediterranean, between 30°N and 45°N, by increasing both winter and summer precipitation. Furthermore, pollen-inferred precipitation values from Lake Trifoglietti and Lake Pergusa in southern Italy also point to significant winter
and summer precipitation during the humid phases of the Early to Mid-Holocene at latitudes south of 40°N (Peyron et al., 2013). During the changeover between the Middle and Late Holocene, the change in orbital parameters, and the weakening of the summer insolation, would, therefore, have led to a weakening of winter and summer storm tracks and more pronounced aridity over the southern Mediterranean, coeval with the end of the AHP.

### 5.2.2 Holocene Centennial-scale climate events in southern Tunisia since 8 ka

The pollen record of the Sebkha Boujmel highlights eight aridity pulses that punctuate the Middle and Late Holocene. These dry events, numbered BJ8 to BJ1, occur at ca. 8 – 7.75, 7 – 6.5, 5.75 – 5.5, 4.9 – 4.6, 3.7 – 3.25, 3 – 2, 1.4 – 1.1 and 0.4 – 0 ka respectively and alternate with seven more humid phases (Fig. 4). The magnitude of the BJ8 arid event, at the base of the sequence, between ca. 8 and 7.75 ka, could be associated with the 8.2 global event (Alley et al., 1997; Mayewski et al., 2004) and a significant instance of aridity recorded in the Mediterranean and in sites at the same latitude as the Sebkha
Boujmel, such as Sebkha Mellala (Gibert et al., 1990). Given the uncertainty regarding [14]C dates at the bottom of the sequence, BJ8 could be contemporaneous with the arid event recorded in various cores from the western (MD95-2043) and central (MD 04-2797 and MD 90-917) Mediterranean, around 7.5 to 7 ka (Fletcher et al., 2012; Combourieu-Nebout et al., 2013; Desprat et al., 2013).

The four subsequent arid phases (BJ7 to BJ4) may be connected with available data from further south in the Libyan Jeffara
(Giraudi et al., 2013) and in the Sahara (Cremaschi et al., 2006; Mercuri, 2008; Cremaschi et al., 2014), particularly with the end of the AHP (Fig. 4). These events may also be linked to events recorded in the Mediterranean. In fact, numerous pollen records from the Mediterranean identify recurrent arid phases which are contemporaneaous with North Atlantic Cooling events (NAC) (Bond et al., 1997; Bond et al., 2001; Wanner et al., 2011). These dry episodes occur during the regression of



Mediterranean plant groups and the increase in semi-desert associations in north-eastern Tunisia (Desprat et al., 2013), and are also evidenced by the decline of deciduous *Quercus* forest in Italy and Greece (Kotthoff et al., 2008a, 2008b; Schmiedl et al., 2010; Combourieu-Nebout et al., 2013) and the western Mediterranean (Combourieu Nebout et al., 2009; Fletcher et al., 2010; Fletcher et al., 2012; Fletcher and Zielhofer, 2013). Such vegetation dynamics may be linked to colder and/or more arid climate (Combourieu Nebout et al., 2009; Combourieu-Nebout et al., 2013; Desprat et al., 2013).

The 4.2 ka event, separating the Middle and Late Holocene (Walker et al., 2012), is seen to be associated with the NAC 3 event, which is evident on the northern side of the Mediterranean basin (Magny et al., 2009; Peyron et al., 2011), in the Siculo-Tunisian Strait (Desprat et al., 2013), in the Medjerda Valley (Faust et al., 2004; Zielhofer et al., 2004) and in the speleothems of Gueldaman Cave (Ruan et al., 2016). However, evidence for this event is not very clear in the Sebkha Boujmel record. Moreover, the 4.2 ka event is not recorded in core sites ODP 976 and MD95 2043 (Fig. 4), nor is it recorded in the eastern Mediterranean (Finné et al., 2011). The last three arid episodes recorded in the Sebkha Boujmel sediments (BJ3, BJ2 and BJ1) are well correlated with the Late Holocene NAC 2, NAC 1, and NAC 0 events, respectively (Fig. 4).

The centennial-scale climatic events recorded in southern Tunisia from 8 ka onwards, may indicate a climatic coupling between the southern Mediterranean and the Sahara during the Middle Holocene. Subsequently, in the Late Holocene, and from 3 ka onwards, atmospheric coupling was established with the North Atlantic. Significant changes in orbital-, solar-, and ice-sheet climate forcing, as well as significant reorganisation of the global climate system and atmospheric circulation at the time of the Mid- to Late Holocene transition, might explain these modifications (Debret et al., 2009; Magny et al., 2013).

## 5.3 Late Holocene landscape evolution and human impact

Human impact has often been considered as overriding during the Holocene, shaping Mediterranean and Saharan landscapes under the effects of agriculture, pastoralism and vegetation clearance (Pons and Quézel, 1998; Schulz et al., 2009). In southern Tunisia, therefore, the vegetation landscape would be the result of significant human impact during the historical period, characterized by the degradation of previously more diverse natural vegetation, consisting of Mediterranean shrubby forest accompanied by tropical steppe with *Acacia raddiana* (Le Houérou, 1959, 1969; Frankenberg, 1986).

In arid environments, it is still difficult to distinguish between human and climatic factors solely on the basis of pollen analysis because several of the marker taxa for anthropisation belong to the natural vegetation of arid regions (Horowitz, 1992). However, in central and southern Tunisia, studies of the dynamics of pre-desert vegetation relevant to human activity, particularly pastoralism, allows nitrophilous and unpalatable taxa to be distinguished within the regional vegetation (Le Houérou, 1959, 1980; Tarhouni et al., 2010; Gamoun, 2014). Crossing the nitrophilous and unpalatable taxa with cultivated and introduced taxa allows the completion of the picture and the interpretation of anthropogenic indicators (Fig. 5).

### 5.3.1 Landscape origin and human impact

The Epipalaeolithic Capsian culture developed in the Maghreb during the Holocene, between ca. 10 and 7 ka, and was replaced by the 'Neolithic of Capsian Tradition' (NCT) around 7 ka (Jackes and Lubell, 2008; Mulazzani, 2013). These





cultural entities are largely distinguished on the basis of typological analysis of their lithic industries. However, in the Maghreb, the history of neolithisation and the emergence of a production economy are still not well understood. To date, very few studies have provided information on the anthropological and economic aspects of Holocene societies in the Maghreb, and for human/environment interactions in this important period of cultural transition (Lubell et al., 1976; Roubet, 2003; Jackes and Lubell, 2008; Mulazzani, 2013). In Capéletti Cave (Aurès, Algeria), the NCT appeared around 7 ka, associated with a pastoral economy based on the rearing of sheep and goats (Roubet, 2003). Archaeological data from the Jeffara Plain and the Jebel Gharbi (Libya) attest to the development of the NCT from 8 ka, with pastoral encampments, typical NCT lithic industries, and ceramics (Barich et al., 2006; Lucarini, 2013; Barich, 2014). Evidence suggests continuity in pastoral activities and transhumance in an area stretching from the foothills of Jebel Gharbi to the Jeffara Plain and on to the coast, up until the historical period (Lucarini, 2013). In the environs of the Sebkha Boujmel, rammadyats (open air sites), although damaged by recent human activity, are still visible and attest to Capsian and/or Neolithic occupation. The Holocene occupation of the region is also evident in the environs of the nearby Sebkha El Melah (Fig.1c) where sites attributed to the NTC are dated to the 7th milennium cal BP (Perthuisot, 1975). Therefore, it appears that ecological niches formed by the paralic sebkhas were occupied and exploited by prehistoric communities during the Holocene period in central Tunisia. These communities hunted and exploited the ecosystems of the lagoons and coasts (Chenorkian et al., 2002; Zoughlami, 2009; Mulazzani, 2013).

At the Sebkha Boujmel, human impact becomes evident after 3 ka with the significant and ongoing appearance of nitrophilous taxa linked to pastoral practices (Fig. 5). No significant anthropogenic impact deriving from agricultural and/or pastoral activities has been recorded previously in the local ecosystems. At that time, the prehistoric societies had only a weak impact on the local environment. The emergence of present-day landscapes can, therefore, be correlated with important climate changes that occurred during the Mid- to Late Holocene transition. In fact, the presence of Mediterranean woodland vegetation in southern Tunisia is attested to at an early date, i.e. around 5 ka, on the island of Djerba (Damblon and Vanden Berghen, 1993). Mediterranean woody species are well represented between 8 and 5.5 ka in the pollen record from the Sebkha Boujmel (Fig. 3). Such woodland formations are indicative of more humid climatic episodes up until 5.5 ka and the AHP termination, before the onset of increasing aridity. Therefore, in the Late Holocene and during the historical period, the vegetation landscape of southern Tunisia was already a steppic- and semi-desert formation in which woodland formations were limited. The same vegetation structure can be observed in pollen data spanning the last two millennia from the nearby Sebkha Mhabeul (Salzmann and Schulz, 1995; Marquer et al., 2008). The dynamic of vegetation associations revealed at the Sebkha Boujmel before 3 ka, leaves no doubt regarding the primary role played by climate in the environmental changes that occurred during the Middle and Late Holocene. Nonetheless, an anthropogenic impact may have amplified the environmental changes of climatic origin during this period.





### 5.3.2 Landscape evolution during historical periods

Human impact increased over the last three centuries BC, during the historical period, with the continuous presence of nitrophilous taxa being recorded and testifying to the permanence of pastoral activities (Fig. 5). From this time onwards, the impact of agricultural activity becomes evident in the record with the addition of cultivated taxa, primarily *Olea* (Fig. 5).

During the Libyo-Phoenician period, since at least the 5th century BC, there is evidence for a high level of human occupation with several urban settlements in the immediate environs of the Sebkha Boujmel and the Bibane lagoon (Fig. 1b) (Trousset and Paskoff, 1991; Drine, 1993; Mattingly, 1995). While the Periplus of Pseudo-Scylax attests to the management of wild olive trees on the nearby island of Djerba as early as the 4th century BC (Shipley, 2011), our data indicate actual cultivation of *Olea* on a large scale from the 3rd century BC onwards. Furthermore, the record for *Vitis* between the 2nd century BC and the 2nd century AD (Fig. 5) can be related to widespread cultivation of vines, the pollen of which does not disperse widely. Notwithstanding the arid and pre-desert environment, *Vitis* could have played an important role in the regional economy, particularly in the mountainous regions and in oases where suitable irrigation techniques would have allowed its development. Seed remains from the Garamantes oases in Fezzan (Libyan Sahara) attest to the cultivation of vines from the beginning of the 1st millennium BC (van der Veen, 1992; Mattingly et al., 2003), well before the consumption and/or introduction of other Mediterranean fruit crops (*Punica granatum* and *Olea europea*) towards the end of the 1st millennium BC (Pelling, 2005).

In the surroundings of the Sebkha Boujmel, the cultivation of olive trees remained relatively important throughout the first two centuries AD. However, a gradual decline in its cultivation is recorded from the beginning of the Roman period (Fig. 5). These changes may be related to the progressive decline in the economic and administrative importance of urban centres close to the Sebkha Boujmel during the Roman period (46 BC – 429 AD), for example the cities of *Zitha* and *Gergis* (Mattingly, 1995). Furthermore, Roman oil presses have been discovered which display technical particularities that attest to low levels of local oil production and modest olive cultivation (Mrabet, 2011). From the 3rd century AD onwards, during the Vandal, Byzantine and Early Medieval periods, including the Arab conquest (from 670 AD), low percentages for cultivated plants indicate that agricultural activity was probably confined to the areas around Berber villages in the mountainous hinterland and that the lowlands were once again dominated by a pastoral economy.

The development of agricultural activities in south-eastern Tunisia between the 3rd century BC and the 2nd century AD coincides, in part, with a slightly more humid period which is recorded in the pollen and clay data between ca. 2 and 1.5 ka (50 BC – 450 AD) (Fig. 4). This episode, commonly known as the Roman Warm Period (RWP), is also recorded at other sites. At the Sebkha Mhabeul, for example, the RWP is associated with a phase of hydrological stability which ends around 430 AD (Marquer et al., 2008). In the Medjerda Valley, the RWP coincides with a phase of stability and soil formation up until 1.7 ka (Faust et al., 2004). Further east, in the Dead Sea, the RWP is associated with an increase in precipitation (Neumann et al., 2010). A surge in agriculture associated with slightly more humid climate is also documented during the



Roman period in other Mediterranean desert margins such as north-western Libya (Gilbertson et al., 1996), in Cyrenaica (Hunt et al., 2002), and at Wadi Faynan in southern Jordan (El-Rishi et al., 2007; Hunt et al., 2007).

The data from Sebkha Boujmel indicates a return to arid climate between around 1.4 and 1.1 ka (BJ2: 550 – 850 AD), contemporary with the Dark Ages (DA) (Fig. 4). This phase is also marked by unstable climatic conditions, associated with a rise in the frequency and intensity of flood events from 550 AD onwards and of arid conditions up to 950 AD at the Sebkha Mhabeul (Marquer et al., 2008) and by a renewal of river activity in the Medjerda Valley (Faust et al., 2004).

Between about 1.1 and 0.5 ka (850 – 1459 AD), the W/D ratio, as well as a drop in herbaceous desert plants, indicate a humid period in southern Tunisia, which, while of modest magnitude, was more important than that recorded during the RWP (Fig. 4). Associated with these wetter conditions, a significant peak in *Artemisia* occurs between 1.1 and 0.8 ka (850 – 1150 AD) (Tab. 2, LPZ5 and Fig. 5), which essentially corresponds to the Medieval Climate Anomaly (MCA). However, it is still difficult to interpret these very high *Artemisia* values, which also feature in the pollen data for the Gulf of Gabes (Brun and Rouvillois-Brigol, 1985), as solely reflecting a change in climate. In fact, during the various humid/arid episodes identified between 8 and 4 ka, Mediterranean and desert taxa fluctuate significantly, but not wormwood (Fig. 3 and Fig. 5). In addition, *Artimesia* does not develop in the Sahara (Ritchie et al., 1985; Ritchie and Haynes, 1987; Lézine et al., 2011), nor on the island of Djerba (Damblon and Vanden Berghen, 1993) during the Early and Middle Holocene. Therefore, *Artemisia* had only a limited presence in the vegetation groups of the southern Tunisian desert margins, which instead were dominated by grassy steppes and which were in direct contact with desert ecosystems (Amaranthaceae) over the course of the Early and Middle Holocene. In the same period, *Artemisia* was well developed further north in central Tunisia (Lebreton and Jaouadi, 2013). Its significant development, associated with that of nitrophilous taxa in southern Tunisia during the Late Holocene, was probably, therefore, linked to human activity and, in particular, to pastoralism. In fact, with the exception of *Artemisia herba alba*, the other species of *Artemisia*, such as *A. campestris*, are not particularly palatable and are avoided by grazing animals (Le Houérou, 1980; Tarhouni et al., 2010; Gamoun, 2014). Furthermore, historically this period (LPZ5, 850 – 1150 AD) corresponds to a period of climate instability with a succession of droughts and famines inducing a decline in the urban and economic structure of the Maghreb. A substantial shift towards pastoralism was also favoured by the migration of nomadic Banu Hilal tribes from southern Egypt to North Africa during the 11[th] century AD (Allaoua, 2003). Thus, at the Sebkha Boujmel, the development of *Artemisia* during the Late Holocene, and particularly between 1.1 and 0.8 ka, may have resulted from a combination of several factors including deterioration in plant diversity under the pressure of intense pastoral activity and an unstable climate with significant seasonal and/or multi-year contrasts.

The humid period recorded at Sebkha Boujmel between ca. 1.1 and 0.5 ka (850 – 1450 AD), and associated with the MCA, is followed between ca. 450 and 95 cal yr BP (BJ1, 1500 – 1855 AD) by a drop in the W/D ratio and the progression of desert taxa (Fig. 4). Arid conditions are re-established in southern Tunisia and are contemporary with the Little Ice Age (LIA, ca. 1450 – 1850). This arid climatic episode during the LIA coincides with an increase in clastic deposits at the top of the sedimentary fill from the nearby Sebkha Mhabeul (Marquer et al., 2008), the renewal of river activity in the Medjerda Valley (Faust et al., 2004), and is also contemporary with the most recent Bond Event (NAC 0) at around 0.4 ka (Fig. 4).





This succession of Wet/Dry climatic conditions, which occurred during the MCA/LIA transition in southern Tunisia, is also evident in other palaeoecological records from around the Mediterranean basin, particularly in sedimentary sequences from caves in Cyrenaica (Hunt et al., 2010; Hunt et al., 2011) and Wadi Faynan in Jordan (Hunt et al., 2007), in pollen data from the coast of northern Syria (Kaniewski et al., 2011) and from the Dead Sea (Neumann et al., 2010), and Siles Lake in

southern Spain (Carrión, 2002a). However, in the western and central Mediterranean, several proxies reveal a reversed succession of climatic conditions, with an arid climate during the MCA and more humid conditions during the LIA. In the central Mediterranean, this climate trend is recorded in the pollen data from Lake Pergusa in Sicily (Sadori et al., 2013) and at Lago di Venere on the island of Pantelleria (Calò et al., 2013). For the western Mediterranean, these climatic conditions (i.e. arid during the MCA and humid during the LIA) are evident in dendroclimatological data obtained from *Cedrus*

*atlantica* in northern Morocco (Esper et al., 2007), in multi-proxy data from Lake Zoñar in southern Spain (Martín-Puertas et al., 2008), and in marine cores from the Alboran Sea (Nieto-Moreno et al., 2013).

In fact, over the course of the last millennium, the MCA (900 – 1350 AD) and the LIA (1500 – 1850 AD) climatic episodes exhibit significant anomalies in temperature and atmospheric circulation in the northern hemisphere which could have been triggered by changes in solar irradiance (Mann et al., 2009; Graham et al., 2011; Swingedouw et al., 2011). Data from

Morocco and Spain indicate a strong correlation between arid/humid variability and positive/negative modes of the NAO (Esper et al., 2007; Martín-Puertas et al., 2008; Trouet et al., 2009; Nieto-Moreno et al., 2013; Wassenburg et al., 2013). Over the course of the MCA/LIA, the marked regionalisation as well as the divergent responses between the central and western Mediterranean on the one hand, and the southern and eastern Mediterranean on the other, may reflect the greater impact of the NAO on storm tracks in the northern and western Mediterranean (Trigo et al., 2004;Trouet et al., 2009). The

Mediterranean expression of the NAO, the Mediterranean Oscilliation (MO), could have driven the contrasts revealed in the palaeoecological records because it would have generated pressure and hydrological contrasts between north-western and south-eastern Mediterranean areas (Dünkeloh and Jacobeit, 2003).

### 5.3.3 Current landscape degradation and desertification processes

The past century has witnessed profound change in the regional ecosystems of southern Tunisia (Fig. 5). Desert taxa are

proliferating around the Sebkha Boujmel, as are other anthropogenic indicators which reflect the increasing impact of human activities on the environment (Fig. 5). Thus, for example, recently introduced xenophytic taxa, such as *Casuarina*, *Acacia cyanophylla* and *Eucalyptus* are present in the Sebkha Boujmel pollen record. Olive growing is becoming more and more developed, and the rise in *Olea* is paralleled by the records for other cultivated taxa (*Phœnix* and *Cerealia*) (Fig. 5). Pastoralism is indicated by ruderal and nitrophilous taxa (*Peganum*, *Polygonum*) and by the more common occurrence of the

pollen taxa *Tamarix*, *Thymelaea* and those of the Zygophyllaceae family *(Nitraria, Fagonia* and *Zygophillum*) (Fig. 3 and 5). The latter plants, which usually show a preference for desert conditions, are unpalatable and resistant to grazing (Le Houérou, 1980; Tarhouni et al., 2010; Gamoun, 2014). Their development, therefore, reflects deterioration in vegetation cover due to overgrazing, which actually favours their spread. Radical changes in the relationships between humans and the



environment are thus recorded in south-eastern Tunisia, particularly in the pollen sequence from the Sebkha Boujmel. In the Jeffara coastal plain, the traditional economy up until the end of the 19th century was based on extensive pastoralism, ensuring the mobility of people and their herds. At this time, the anthroposystem was relatively balanced in the context of the limited, fragile resources (Talbi, 1997). After 1881 AD, under the French protectorate, these traditional social and production

structures were supressed in favour of increased sedentism and the introduction of new systems of agricultural production based on individual ownership of land and intensive olive and cereal production (Abaab, 1986). These economic and social changes resulted in increased impacts on the environment (Talbi et al., 2009). In fact, the disappearance of collective herd management led to a reduction in herd movements and caused locally intensive overgrazing with rapid deterioration of vegetation cover accompanied by an increase in unpalatable substitute species. All of these changes led to an accentuation of

wind and water erosion and an intensification of the process of desertification (Sghaier et al., 2012).

## 6 Conclusions

Pollen and clay mineralogical data from the Sebkha Boujmel underline the significant potential of salt-flat sedimentary records as palaeoecological archives. Sebkhas are particularly conducive to the preservation of pollen in arid regions and, therefore, constitute a rare sedimentary resource favourable to palynology in sub-arid environments and allowing the

reconstruction of the dynamics of vegetal environments. Pollen data from the Sebkha Boujmel provide the first, continuous, high-resolution record to allow the detailed reconstruction of the landscape dynamics of south-eastern Tunisia over the last eight millennia. The palaeoecological records from the Sebkha Boujmel tie in with other regional and extra-regional proxies and the impact of global climate changes can be superimposed on local conditions with various degrees of sensitivity. These events highlight the complexity of climatic trajectories in the south-central Mediterranean, with the influence of several

closely correlated parameters, such as solar forcing and NAO-type circulation, from ca. 8000 cal BP onwards. Furthermore, the vegetation dynamic suggests that human activity could have amplified the arid climatic event which began in the Late Holocene. The anthropogenic impact became predominant when a sedentary way of life developed, i.e. during Classical Antiquity and, later, under the French protectorate. In the case of the latter period, spanning the last century, anthropogenic changes become evident and have led to profound alterations to the spatial organisation of the territory. The pronounced

development of desert taxa and markers of agricultural and pastoral activity, attest to fundamental changes in the relationships between humans and the environment in southern Tunisia at the present time. The joint impact of climate change and human pressure result in a drastic deterioration of the landscape by accentuating the process of desertification.

## Acknowledgements

The authors wish to thank the Institut National du Patrimoine de Tunis (INP), which organised the fieldwork and provided

necessary authorisation and logistical support for coring and for the archaeological survey of the environs of the Sebkha



Boujmel. This research was financially supported by the European Commission within the framework of the Erasmus Mundus International Doctorate in Quaternary and Prehistory (IDQP) coordinated by the Università degli studi di Ferrara (Italy). This work was part-funded by the Tunisian Minister for Higher Education and Scientific Research and by the Mistrals-PaleoMex programme.

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





**Table 1:** Conventional AMS-radiocarbon dates, reservoir corrected and 2σ range calibrated ages from Sebkha Boujmel (BJM2 core).

| Lab. code | Depth (cm) | Conventional $^{14}C$ age | error | Reservoir corrected age | Lower cal. age | Upper cal. age | Mean age cal yr BP | 2 σ Error |
|---|---|---|---|---|---|---|---|---|
| BETA 403124 | 40 | 1570 | 30 | 1570 | 1395 | 1533 | 1464 | 98 |
| BETA 403123 | 75 | 2600 | 30 | 2200 | 2140 | 2315 | 2228 | 124 |
| BETA 384850 | 110 | 3450 | 30 | 3050 | 3174 | 3350 | 3262 | 124 |
| BETA 403122 | 120 | 3750 | 30 | 3350 | 3545 | 3643 | 3594 | 69 |
| POZ-75206 | 127 | 4305 | 35 | 3905 | 4237 | 4425 | 4331 | 133 |
| BETA 413550 | 134 | 5170 | 30 | 4770 | 5465 | 5589 | 5527 | 88 |
| POZ-75205 | 141 | 5985 | 35 | 5585 | 6300 | 6414 | 6357 | 81 |
| BETA 413551 | 146 | 5300 | 30 | 4900 | 5588 | 5663 | 5626 | 53 |
| BETA 403121 | 150 | 6640 | 30 | 6240 | 7153 | 7254 | 7204 | 71 |
| BETA 413552 | 156 | 7510 | 30 | 7110 | 7922 | 8000 | 7961 | 55 |
| BETA 384851 | 160 | 6370 | 30 | 5970 | 6730 | 6891 | 6811 | 114 |



**Table 2:** Local Pollen Zones, vegetation and inferred climatic and/or anthropogenic changes based on the pollen record from core BJM2.

| Local Pollen Zone (LPZ) | Interval (cm) | Age (Cal yr BP) | LPZ description | Vegetation changes | Inferred climate change and/or anthropogenic impact |
|---|---|---|---|---|---|
| **LPZ-7** | 0-3 | (-63) - 95 | Rise in cultivated (mainly *Olea*), nitrophilous and desert taxa. Introduced taxa are recorded. High values of *Artemisia* and decrease in Poaceae. | Pre-desert steppes, degraded by agriculture and pastoralism | Environment greatly modified and degraded by humans |
| **LPZ-6** | 3-21 | 95-827 | *Artemisia*, Fabaceae, *Ephedra* cultivated and nitrophilous taxa decrease. Increase in Cichorioideae, Poaceae and some desert taxa. *Cornulaca-Traganum* type, Brassicacaeae and Caryphyllaceae remain scarce. | Mixed steppes with *Artemisia* and Poaceae | More humid climate evolving towards aridity with limited anthropic impact |
| **LPZ-5** | 21-31 | 827-1126 | Very high values of *Artemisia* (ca. 45%) and increase in *Ephedra*, Fabaceae, cultivated and nitrophilous taxa. Decrease in *Cornulaca-Traganum* type, Amaranthaceae, Asteraceae, Brassicaceae and Caryophyllaceae. | Pre-desert steppes with *Artemisia*, agriculture and pastoralism | Slightly humid climate with significant anthropic impact (pastoralism) |
| **LPZ-4** | 31-71 | 1126-2026 | Increase in *Artemisia*, *Ephedra*, Asteraceae Asterioideae. Increase in cultivated taxa (*Olea* + *Vitis*) and nitrophilous. High values for *Cornulaca-Traganum* type and desert taxa. | Pre-desert steppes | Arid climate with significant anthropic impact (agriculture and pastoralism) |
| **LPZ-3** | 71-104 | 2026-3122 | Significant decrease in Poaceae and *Rumex*. Evident increase in *Artemisia, Cornulaca-Traganum*-type, steppe and desert taxa. First occurence of cultivated taxa at ca 2.3 ka. | Pre-desert steppes with *Artemisia* | Arid climate with limited anthropic impact |
| **LPZ-2** | 104-147 | 3122-6366 | Increase in *Pistacia*, *Pinus*, *Juniperus*, Poacea and aquatics. Decrease in *Cornulaca-Traganum* type and Caryophyllaceae. Low percentages of steppe and desert herbaceous taxa. | Mediterranean mattoral and grass steppes, progressively evolving towards pre-desert steppes | Humid climate progressively evolving towards arid climate |
| **LPZ-1** | 147-160 | 6366-7982 | Relatively high values for *Pistacia*, *Juniperus*, *Quercus* type-*ilex*. Significant percentages of Asteraceae, Poaceae and *Cornulaca-Traganum* type. Low values of aquatic taxa and A*rtemisia*. | Grass and desert steppes and Mediterranean mattoral | Arid climate with relative moisture in favourable biotopes |



**Figure 1:** Geographical settings. **a**: Large map of the Mediterranean Basin showing the location of the study area (outlined in red, inset maps b and c) and of selected Mediterranean records quoted in the text. CR: Chott Rharsa, GC: Gueldaman Cave, GG: Gulf of Gabes, HM: Hassi el Mejnah, JG: Jbel Gharbi, JR: Jerba Island, LP: Lake Preola, LS: Lake Siles, MD04: core MD04-2797CQ, MD90: core MD90 917, MD95: core MD95 2043, MV: Medjerda Vally, ODP: ODP site 976, SHM: Sebkha Halk el Menjel, SK: Sebkha Kalbiyya, SM: Sebkha Mellala, WT: Wadi Teshuinat. **b**: Inset map that shows main regional setting for Sebkha Boujmel including topography, vegetation (after Gammar, 2008), hydrographic network, archaeological sites (after Drine, 1993) and current cities. Vegetation 1: Halophilous vegetation. 2: Gypsophilous steppe with *Lygeum* and *Zygophyllum*. 3: Cultivated fields (*Olea*) and steppe with *Rhanterium*, *Holoxylon scoparium* and *Artemisia*. 4: Steppe with *Holoxylon scoparium* and *Holoxylon schimittianum*. 5: Steppe with *Stipa tenacessima* and matorral with *Juniperus phoenicea*. 6: Desert vegetation with *Helianthemum* sp., *Calligonum*, *Gymnocarpos decander*, *Traganum nudatum*. **c**: Detailed map displays the location of Sebkha Boujmel, the studied core (BJM2) and other nearby sebkhas mentioned in the text.





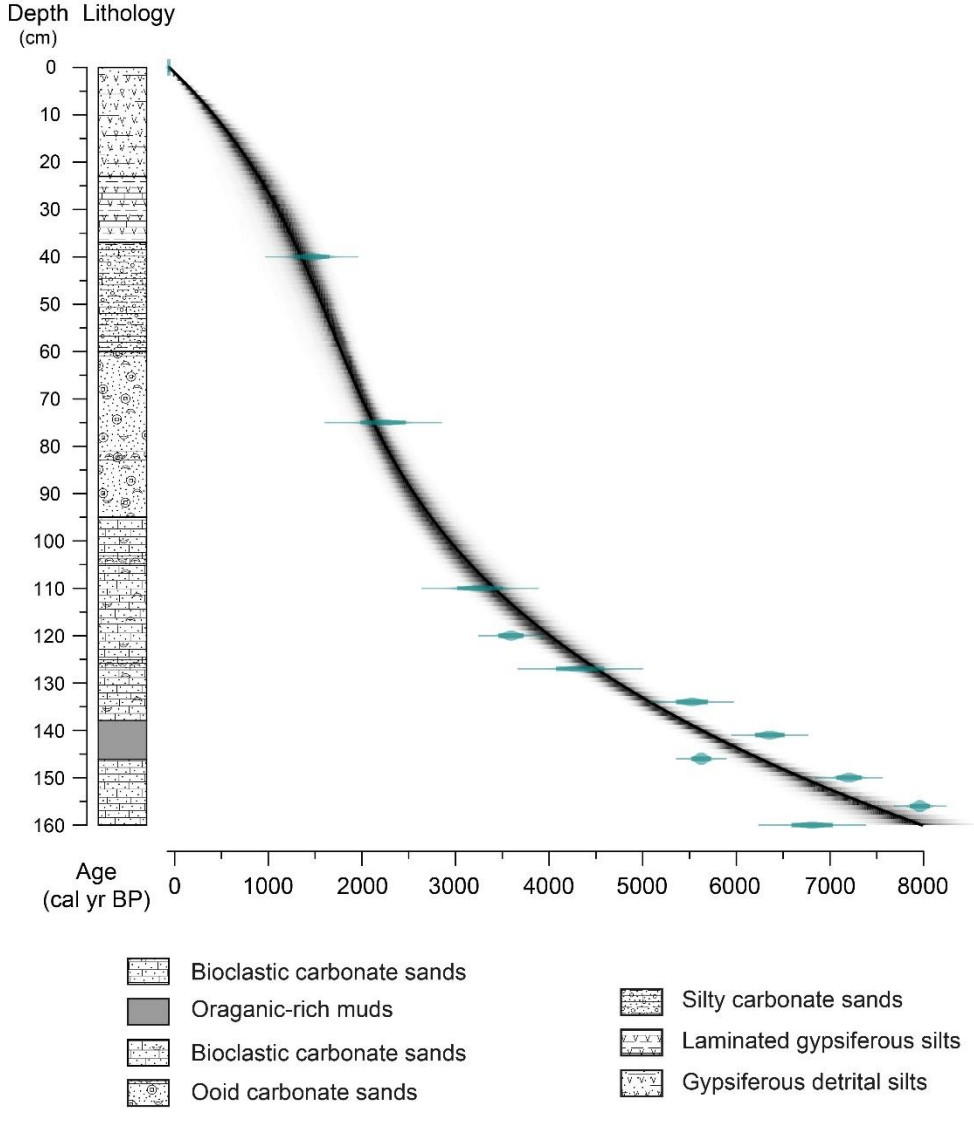

**Figure 2:** Lithology and age-depth model for BJM2 core. The age-model was obtained by 3$^{rd}$ degree polynomial regression with 10k model iteration using the package Clam 2.2 (Blaauw, 2010). Grey scales show the distribution of all age-models at 500 equally spaced core depths.





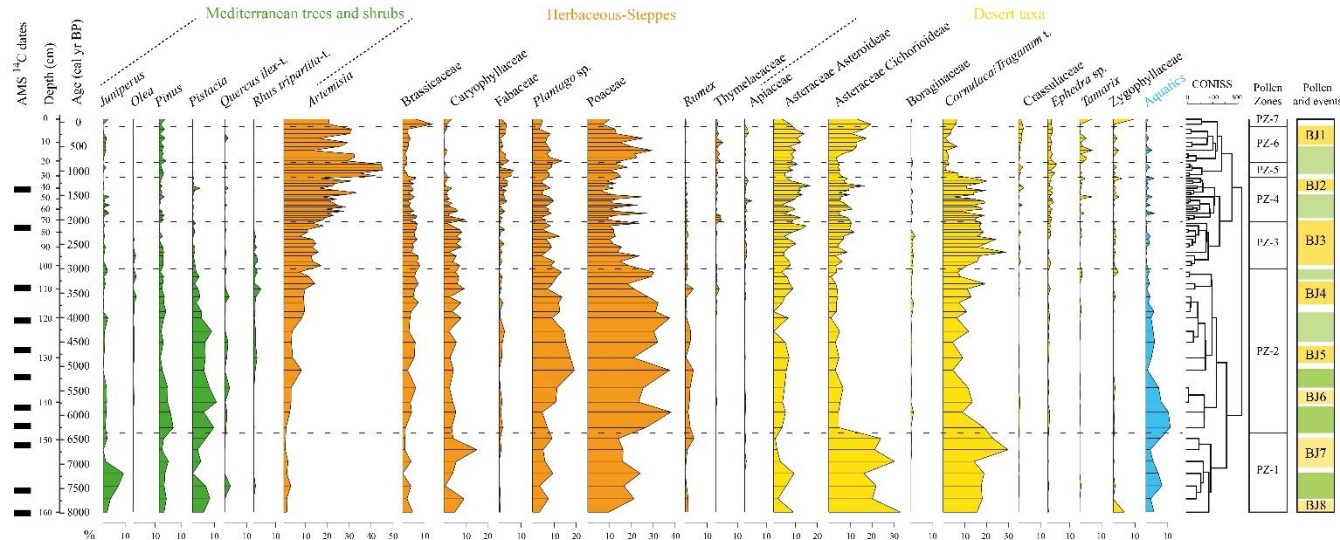

**Figure 3:** Pollen pourcentage diagram from Sebkha Boujmel showing the evolution of the main pollen taxa of the tree ecological groups (Mediterranean, steppe and desert taxa) plotted against age. Pollen taxa and types from the same genus or family and with the same ecology are grouped on the pollen diagram including Boraginaceae (*Moltkia ciliata*, *Onosma* and *Echium*), *Ephedra* sp. (*Ephedra fragilis*-t. and *Ephedra distachia*-t.) and Zygophyllaceae (*Fagonia*, *Nitraria* and *Zygophillum*). Pourcentage of acquatics pollen are calculated based on the total sum of pollen grains identified in each pollen spectrum. Black boxes on the depth scale indicate positions of [14]C AMS dates.

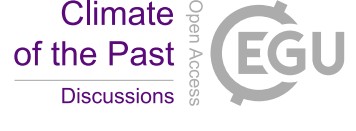



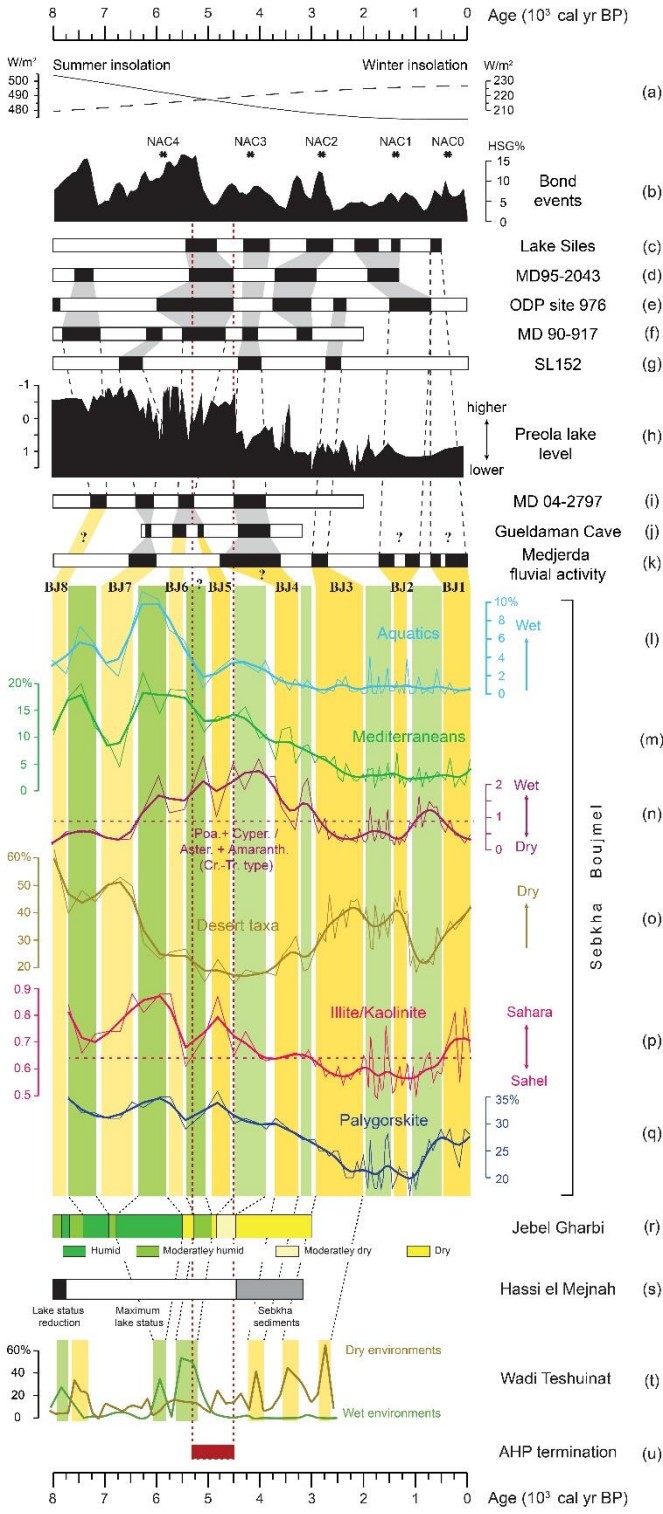





**Figure 4:** Comparaison of pollen and clay mineralogy data from BJM2 core with paleoclimate records from the Mediterranean and the Sahara. **a)** Summer (June mid-month) and winter (December mid-month) insolation (W/m$^2$) curve at 30° N (Berger and Loutre, 1991). **b)** Drift Ice Record from the North Atlantic: Stacked % of HSG (Hematite Stained Grains) of 4 records with stars indicate the North Atlantic Cooling events (Bond et al., 2001; Bond et al., 1997). **c)** Desiccation events at Lake Siles (southern Spain, 38°24'N, 2°30'W) based on *Pseudoschizaea* microfossil abundance (Carrión, 2002a). **d)** Episodes of Mediterranean forest decline based on pollen data from MD95-2043 core in the Alborean Sea (36°9'N, 2°37'W) (Fletcher et al., 2012). **e)** Temperate forest decline events ODP site 976 core in the Alboran Sea (36°12'N, 4°18'W) (Combourieu Nebout et al., 2009). **f)** Cold/dry events on core MD 90-917 (41°N, 17°37'E) pollen record from the Adriatique Sea (Combourieu-Nebout et al., 2013). **g)** Arid events from SL152 core (Aegean sea, 40°19′N, 24°65′E) (Kotthoff et al., 2008a, 2008b; Schmiedl et al., 2010). **h)** Lake-level fluctuations (CA scores) at Lago Preola in Sicily (37°37'N, 12°38'E) (Magny et al., 2011). **i)** Identified arid spells on pollen data from MD 04-2797 core in the Siculo-Tunisian Strait (36°57'N,11°40'E) (Desprat et al., 2013). **j)** Droughts episodes based on stable isotopes record from stalagmite GLD1-stm4, Gueldaman Cave, N-Algeria, (36°26'N, 4°34'E) (Ruan et al., 2016). **k)** Phases of increased fluvial activity with low soil formation and arid climate in the Medjerda river valley, Northern Tunisia (Faust et al., 2004; Zielhofer et al., 2004). Pollen and clay mineralogy data from sebkha Boujmel (BJM2 core, 33°16'N°, 11°05'E) with bold lines figuring the cubic smoothing splines and dashed lines averages for the I/K and W/D ratios. **l)** percentages of fresh water (Cyperaceae, *Glyceria*, *Juncus*, *Lemna*, *Potamogeton*, *Rumex aquaticus*-t., *Typha-Sparganium*) and **m)** Mediterranean trees and shrubs (*Buxus, Ceratonia, Cistus, Juniperus,* Lamiaceae*, Myrtus, Nerium, Olea,* Papaveraceae*, Pinus, Pistacia, Quercus ilex*-t.*, Quercus* deciduous-t.*, Rhus tripartita*-t.) pollen taxa. **n)** Wet/Dry (W/D) pollen ratio (Poaceae + Cyperaceae / Asteraceae Cichorioideae + Asteraceae Asteroideae + Amaranthaceae *Cornulaca-Traganum* Type). **o)** Percentages of desert pollen taxa (Apiaceae, *Asphodelus*, Asteraceae Asteroideae, Asteraceae Cichorioideae, *Calligonum, Capparis, Cistanche, Cleome, Cornulaca-Traganum*-t., Crassulaceae, Cucurbitaceae, *Echium, Ephedra distachia*-t.*, Ephedra fragilis*-t.*, Fagonia, Helianthimum*, Malvaceae*, Moltkia ciliata*, *Neurada, Nitraria, Onosma, Reaumuria, Tamarix* and *Zygophillum*). **p)** Illite/Kaolinite ratio and **q)** palygorskite pourcentages. **r)** Holocene climatic events in the Libyan Jeffara (Jebel Gharbi -northwestern Libya, 32°N) based on sedimentological and palynological data (Giraudi et al., 2013). **s)** Paleohydrological changes at Hassi el Mejnah in Algeria (31°14'N, 2°30'E) (Gasse, 2002; Hoelzmann et al., 2004). **t)** Pollen data from rock shelters in the Wadi Teshuinat area (Tadrart Acacus Massif, Central Sahara, South-West Libya, 24°30'-26° N, 10-12°E). (Cremaschi et al., 2014; Mercuri, 2008). **u)** Estimated duration of the African Humid Period termination (4.9 ka ± 400, 2σ) as indicated by eolian dust flux data on marine sediment cores from northwest African margin (McGee et al., 2013).



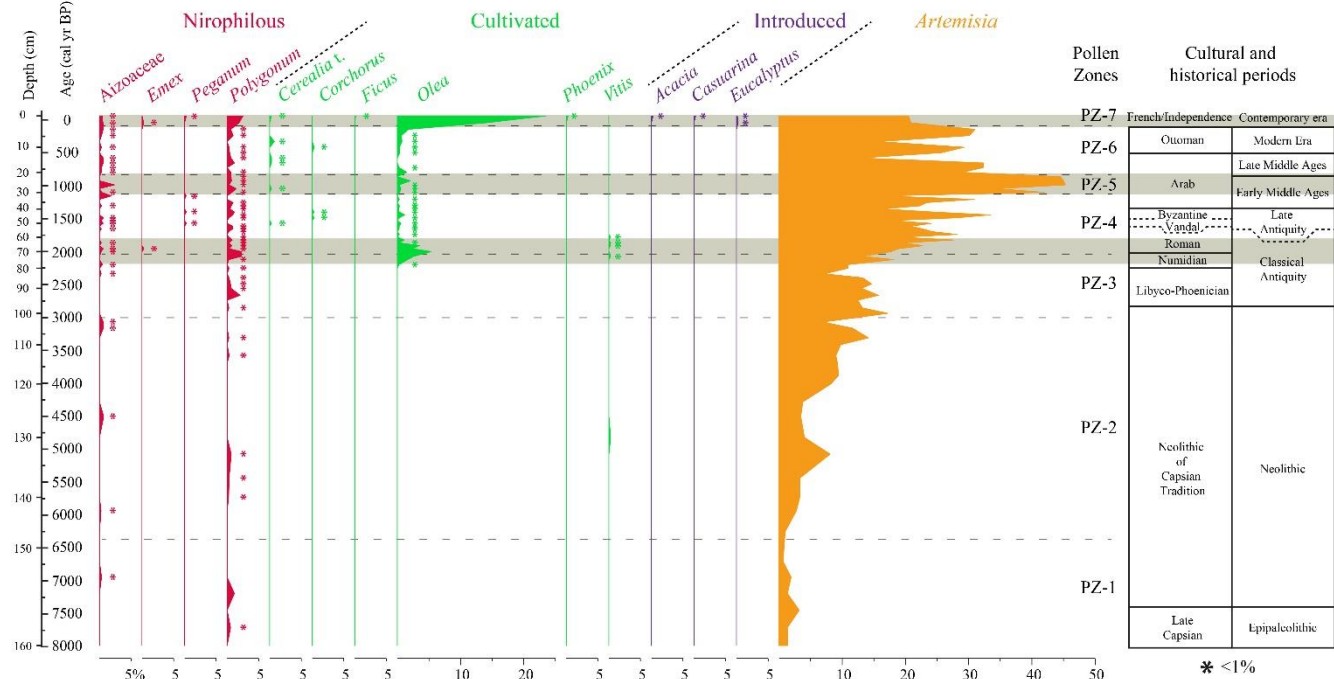

**Figure 5:** Detailed pollen diagram showing the evolution of Artemisia and selected Anthropogenic Pollen Indicators (API) including cultivated (*Cerealia* t., *Corchorus*, *Ficus*, *Olea*, *Phoenix*, *Vitis*), nitrophilous (Aizoaceae, *Emex*, *Peganum*, *Polygonum*) and introduced plants taxa (*Acacia* t. *cyanophilla*, *Casuarina*, *Eucalyptus*). Grey bars highlight periods of enhanced human pastoral and/or agriculture activities. Table on the right side indicates main cultural and historical subdivisions of the Tunisian history.