# Peer review of "Environmental changes, climate and anthropogenic impact in southern-eastern Tunisia during the last 8 kyr"

_Climate of the Past, 2016_

## Referee Comment (RC1) · Anonymous Referee #1 · 1 Apr 2016

\* This is a job of a good scientific level. However, substantive comments are important to report. These remarks concern especially the current vegetation of the study area. Most of these remarks is:

\* The authors say: Vegetation is sparse and adapted to the arid conditions with psammophyte shrubs (Calligonum sp., Ephedra alata subsp. alenda and Retama raetam) and desert herbaceous plants such as Amaranthaceae (Cornulaca monacantha, Traganum nudatum), Boraginaceae (Echium sp., Moltkiopsis ciliata), Zygophyllaceae (Fagonia sp., Nitraria retusa), Brassicaceae (Henophyton deserti) and Euphorbiaceae (Euphorbia guyoniana). Authors must be careful: all these species are no herbaceous, but woody plants.

\* The significant increase in Artemisia (wormwood) between 1.1 and 0.8 ka (850 – 1150 AD) is linked to intensive pastoral activity, associated with heightened interannual and/or seasonal climatic instability. The appearance of Artemisia is newer at the vegetation of southern Tunisia.

Moreover, I invite the authors to read the synthesis the Houérou (1959 & 1969 ), already mentioned in this work and especially Le Houérou (1994). According to The Houérou, the occurrence of Artemisia is very recent, and linked to contemporary and actual human activity. According to this author, as well as all recent studies, the occurrence of Artemisia herba-alba is linked to the actual degradation of the steppe of Alfa, which exists on loamy soils, and Glacis. On the other hand, the appearance of Artemisia campestris is related to actual clearing steppes Rhanterium suaveolens, which exists on sandy substrate of the Djeffara plain of the Tunisian south.

Salvadora persica is a species of the Middle East and the Persian Gulf, and has never existed in North Africa.

Several scientific plant species names are written with errors. example, Haloxylon scoparium not Holoxylon scoparium in the legend to Figure 1.

The authors employ often old scientific nomenclature. I invite them to review the names of species according to the new nomenclature , proposed by Le Floc'h , Boulos & Vela (2010).

Finally, authors should consider these remarks on the current flora to claim the publication of this work.

---

## Referee Comment (RC2) · Anonymous Referee #2 · 4 Apr 2016

General comments

I read with great pleasure this contribution. It is of high scientific value, and exemplifies the need for more studies of marginal, semi-desertic to desertic environments, to complement our regional knowledge of palaeoclimatic changes. Such settings, often neglected, possess a wealth of paleoenvironmental information awaiting researchers. The manuscript is clear, well-constructed and well-written. Particularly appreciable is the fact that the authors clearly explain both their scientific approach and how they treated the data, and take great care to separate results and interpretations; which are carefully formulated. The use of both pollen- and clay minerals-based ratio indicators for the identification of wetter/drier periods is an interesting approach, and may help

other researchers dealing with similar environments.

Specific comments

Radiocarbon dating: The authors state that the organic matter they dated is "of mixed origin, composed of marine planktonic/algal material and continental woody material" (p.5, l.3-4). In this case, one may wonder if correction for marine reservoir effect is appropriate – it may lead to an overcorrection of the dates? It would have been better, of course, to be able to date terrestrial macrofossils instead of bulk organic matter – but these are notoriously missing in such environments. However, for further studies, the authors may consider the possibility of picking small particles of charcoal (200-500 $\mu$m), which may be present in the sediment and may provide more reliable ages. This, and the general aspect of the age-depth model (Fig. 2) indicate that indeed time constrain may not be that reliable below ca. 4000 BP, and this renders it difficult to compare short-term events with other records. Although the authors take great care to emphasis this in the discussion, it may be better to acknowledge this issue by introducing more caution (e.g., by the use of ca.) when summarising the results (e.g. in the abstract).

Comparison to other records: Roman period: there are actually some indications that the climate may have been colder during the Roman period in the Middle East, so the term "Roman Warm Period" may not be appropriate for the whole region... (e.g. Issar 2003 - Climate Changes during the Holocene and their Impact on Hydrological Systems, INTERNATIONAL HYDROLOGY SERIES, Cambridge university press).
* * *

---

## Editor Comment (EC1) · J. Guiot (Editor) · 26 Apr 2016

Dear authors, we have now received two evaluations of your paper. The first reviewer consider that your paper has a good scientific level, but its needs to be more rigorous in the vegetation nomenclature and should be based on more solid reference literature on the Mediterranean vegetation. The second reviewer has appreciated the paper but wish a better consideration of the limits of the dating on bulk samples. Another point is that the Roman period is not usually considered as warm. We expect now that you reply to the reviewers in a cover letter accompanied by the ms revised in track change mode (as annex of the cover letter). Best regards Joel Guiot

---

## Author Comment (AC1) · 18 May 2016

- The authors say: Vegetation is sparse and adapted to the arid conditions with psammophyte shrubs (Calligonum sp., Ephedra alata subsp. alenda and Retama raetam) and desert herbaceous plants such as Amaranthaceae (Cornulaca monacantha, Traganum nudatum), Boraginaceae (Echium sp., Moltkiopsis ciliata), Zygophyllaceae (Fagonia sp., Nitraria retusa), Brassicaceae (Henophyton deserti) and Euphorbiaceae (Euphorbia guyoniana). Authors must be careful: all these species are no herbaceous, but woody plants.

This has been corrected by mentioning only the psammophyte shrubs as forming the main vegetation of the desert. The ecology and plant types of these species follow Le Houérou (1959) and Pottier-Alapetite (1979, 1981).

- The significant increase in Artemisia (wormwood) between 1.1 and 0.8 ka (850 – 1150 AD) is linked to intensive pastoral activity, associated with heightened interannual and/or seasonal climatic instability. The appearance of Artemisia is newer at the vegetation of southern Tunisia. Moreover, I invite the authors to read the synthesis the Houérou (1959 & 1969), already mentioned in this work and especially Le Houérou (1994). According to The Houérou, the occurrence of Artemisia is very recent, and linked to contemporary and actual human activity. According to this author, as well as all recent studies, the occurrence of Artemisia herba-alba is linked to the actual degradation of the steppe of Alfa, which exists on loamy soils, and Glacis. On the other hand, the appearance of Artemisia campestris is related to actual clearing steppes Rhanterium suaveolens, which exists on sandy substrate of the Djeffara plain of the Tunisian south.

We thank the referee for these constructive remarks. Current studies of the dynamics of steppic vegetation associations in Tunisia are important in order to throw light on the Holocene records and to explain certain changes in the vegetation structure such as those observed in fossil pollen spectra. The elements suggested by the referee have been inserted into our discussion on the dynamics of *Artemisia* during the Holocene, particularly in relation to the replacement of the *Rhanterium suaveolens* steppe by the *Artemisia campestris* steppe in the Jeffara (Chaieb and Zaâfouri, 2000; Genin et al. 2006).

However, notwithstanding the complementarity between contemporary botanical studies and palaeoecological data produced by pollen analysis, as for example in the case of *A. campestris* and *Rhanterium suaveolens*, we feel that a significant difference exists in approaching vegetation dynamics within a temporal perspective at the scale of the Holocene through pollen analyses. Thus, for *Artemisia*, the pollen data from Sebkha Boujmel indicate a relatively early and progressive development, closely linked to anthropic activity, even though other factors could also have played a role in this development. These data do not support the claim for an exclusively contemporary and very recent development. On the contrary, it is important to place the recent development of *Artemisia* within a long-term dynamic which is also apparent in other pollen diagrams in Tunisia (Brun, 1983 ; Brun and Rouvillois-Brigol, 1985).

- Salvadora persica is a species of the Middle East and the Persian Gulf, and has never existed in North Africa.

*Salvadora Persica* is mentioned with reference to the work carried out by Giraudi and colleagues (2013) who report the occurrence of pollen of this species in the nearby Libyan Jeffara to the south. This species is currently reported from many Saharan mountains such as Hoggar and Tassili (e.g. Ozenda, 2004, p.366). As regards the Holocene palaeo-botanical records, besides the data from the Libyan Jeffara, both pollen (Mercuri, 2008) and charcoal (Neumann and Uebel, 2001) of *Salvadora persica* are reported from Holocene archaeological sites in the Libyan Sahara.

- Several scientific plant species names are written with errors. example, Haloxylon scoparium not Holoxylon scoparium in the legend to Figure 1. The authors employ often old scientific nomenclature. I invite them to review the names of species according to the new nomenclature, proposed by Le Floc'h, Boulos & Vela (2010). Finally, authors should consider these remarks on the current flora to claim the publication of this work.

All botanical species names in the text and figure captions have been checked for typing mistakes and have been duly corrected in accordance with Le Floc'h et al., 2010. The latter reference has been added to the paper and inserted in the text (P.3, L.25). However, we still refer to *Artemisia herba-alba* (p.15,l.22) in order to make it easier for readers to follow this work in respect to previous published data and studies, and also because the evidence for *Artemisia saharae* is not yet confirmed with certitude in Tunisia (Le Floc'h et al. 2010).

References:

Chaieb, M., and Zaâfouri, M. S.: L'élevage extensif, facteur écologique primordial de la transformation physionomique du cortège floristique en milieu steppique tunisien, in: Rupture : nouveaux enjeux, nouvelles fonctions, nouvelle image de l'élevage sur parcours, edited by: Bourbouze, A., and Qarro, M., CIHEAM, Montpellier, 217-222, 2000.

Le Floc'h, E., Boulos, L., and Vela, E.: Catalogue synonymique commenté de la Flore de Tunisie, Ministère de l'Environnement et du Développement durable - Banque Nationale de Gènes, Montpellier-Tunis, 500 pp., 2010.

Mercuri, A. M.: Human influence, plant landscape evolution and climate inferences from the archaeobotanical records of the Wadi Teshuinat area (Libyan Sahara), Journal of Arid Environments, 72, 1950-1967, http://dx.doi.org/10.1016/j.jaridenv.2008.04.008, 2008.

Neumann, K., and Uebel, D.: The cold Early Holocene in the Acacus: Evidence from charred wood, in: Uan Tabu in the Settlement History of the Libyan Sahara, edited by: Garcea, E. A. A., Arid Zone Archaeology – Monographs, 2, All'Insegna del Giglio, Firenze, 211–213, 2001.

---

## Author Comment (AC2) · 18 May 2016

- Radiocarbon dating: The authors state that the organic matter they dated is "of mixed origin, composed of marine planktonic/algal material and continental woody material" (p.5, l.3-4). In this case, one may wonder if correction for marine reservoir effect is appropriate – it may lead to an overcorrection of the dates? It would have been better, of course, to be able to date terrestrial macrofossils instead of bulk organic matter – but these are notoriously missing in such environments. However, for further studies, the authors may consider the possibility of picking small particles of charcoal (200-500 $\mu$m), which may be present in the sediment and may provide more reliable ages. This, and the general aspect of the age-depth model (Fig. 2) indicate that indeed time constrain may not be that reliable below ca. 4000 BP, and this renders it difficult to compare short-term events with other records. Although the authors take great care to emphasis this in the discussion, it may be better to acknowledge this issue by introducing more caution (e.g., by the use of ca.) when summarising the results (e.g. in the abstract).

Thank you for this remark and the suggestion about dating charcoal particles for future research. The referee's advice has been taken on board by using "ca." for all dates within the abstract and the paper text.

As the referee rightly points out, terrestrial organic material suitable for dating is lacking in such sediments. The presence of benthic foraminifera, the mixed origin of the dated materiel, and the absence of data regarding the relative percentages of planktonic/algal material and continental woody material within each sample, obliged us to correct the dates for marine reservoir effect. In all, we think that we have an acceptable chronological framework as maximum age error ranges from 1 to 2 centuries considering the error bars with or without corrections for reservoir effect. Efforts have also been made to obtain a maximum age control within the sequence by cross-checking [14]C dates and tephrochronological data. Unfortunately, sieving and observation under binocular microscope do not allow tephra particles to be identified, even though such material has been reported from the nearby Sebkha Mhabeul (Marqueur et al.2008).

- Comparison to other records: Roman period: there are actually some indications that the climate may have been colder during the Roman period in the Middle East, so the term "Roman Warm Period" may not be appropriate for the whole region. . . (e.g. Issar 2003 - Climate Changes during the Holocene and their Impact on Hydrological Systems, INTERNATIONAL HYDROLOGY SERIES, Cambridge university press).

We thank the referee for this remark highlighting the problems arising from the use of northern European terminology when dealing with climate change in other regions, especially when this terminology includes climatic indications such as "warm". The "Roman Warm Period" does not have a global signal as an RCC with important regional differences and peculiarities (e.g. Martín-Puertas et al. 2008). Therefore, we use "Roman Warm Period" as a generic term for this climate event and refer to it within the text as a climate episode "commonly known as the Roman Warm Period (RWP)". Based on our data, we do not yet have information as to whether this period was warm or cold and the main element that we discuss is the relative wetness of this period based on our data compared to other records from the southern Mediterranean. Therefore, we have decided to use the term "Roman Humid Period-RHP", as done, for example, by Martín-Puertas et al. 2009 and Jiménez-Moreno et al. 2013 for southern Spain. "Roman Humid Period" seems more appropriate and consistent with our data and other southern Mediterranean records discussed in the text, such as the Middle East where a colder and more humid climate prevailed during the Roman period (Issar, 2003).

References:

Issar, A. S.: Climate Changes during the Holocene and their Impact on Hydrological Systems, Cambridge University Press, 2003.

Jiménez-Moreno, G., García-Alix, A., Hernández-Corbalán, M. D., Anderson, R. S., and Delgado-Huertas, A.: Vegetation, fire, climate and human disturbance history in the southwestern Mediterranean area during the late Holocene, Quaternary Research, 79, 110-122, http://dx.doi.org/10.1016/j.yqres.2012.11.008, 2013.

Martín-Puertas, C., Valero-Garcés, B. L., Brauer, A., Mata, M. P., Delgado-Huertas, A., and Dulski, P.: The Iberian–Roman Humid Period (2600–1600 cal yr BP) in the Zoñar Lake varve record (Andalucía, southern Spain), Quaternary Research, 71, 108-120, http://dx.doi.org/10.1016/j.yqres.2008.10.004, 2009.

---

## Author Response (AR2)

Dear Editor,

On behalf of all co-authors, I would like first to thank the Anonymous Referees and the editor for their constructive and insightful comments. This helps us a lot to improve the paper.

All issues and suggestions have been addressed in the review reports. The corrected manuscript has been then revised following the suggestions of the referees and the editor. Below are listed (i) a point-by-point reply to the reviewer's comments - original comments in black fonts and responses in blue- with the modifications done in the manuscript and (ii) a marked-up manuscript version.

Hoping our revised manuscript will be suitable for publication in Climate of the Past.

Yours sincerely,

Sahbi Jaouadi

**Anonymous Referee #1**

- The authors say: Vegetation is sparse and adapted to the arid conditions with psammophyte shrubs (Calligonum sp., Ephedra alata subsp. alenda and Retama raetam) and desert herbaceous plants such as Amaranthaceae (Cornulaca monacantha, Traganum nudatum), Boraginaceae (Echium sp., Moltkiopsis ciliata), Zygophyllaceae (Fagonia sp., Nitraria retusa), Brassicaceae (Henophyton deserti) and Euphorbiaceae (Euphorbia guyoniana). Authors must be careful: all these species are no herbaceous, but woody plants.
- Author's response: This has been corrected by mentioning only the psammophyte shrubs as forming the main vegetation of the desert. The ecology and plant types of these species follow Le Houérou (1959) and Pottier-Alapetite (1979, 1981).
- Manuscript change: (p. 4, l. 1-2): the sentence is corrected as follows: "Vegetation is sparse and adapted to the arid conditions mainly with psammophyte shrubs such as *Calligonum* sp., *Cornulaca monacantha, Ephedra alata* subsp. *alenda, Moltkiopsis ciliata, Retama raetam* and *Traganum nudatum.*"
- The significant increase in Artemisia (wormwood) between 1.1 and 0.8 ka (850 1150 AD) is linked to intensive pastoral activity, associated with heightened interannual and/or seasonal climatic instability. The appearance of Artemisia is newer at the vegetation of southern Tunisia. Moreover, I invite the authors to read the synthesis the Houérou (1959 & 1969), already mentioned in this work and especially Le Houérou (1994). According to The Houérou, the occurrence of Artemisia is very recent, and linked to contemporary and actual human activity. According to this author, as well as all recent studies, the occurrence of Artemisia herba-alba is linked to the actual degradation of the steppe of Alfa, which exists on loamy soils, and Glacis. On the other hand, the appearance of Artemisia campestris is related to actual clearing steppes Rhanterium suaveolens, which exists on sandy substrate of the Djeffara plain of the Tunisian south.
- Author's response: We thank the referee for these constructive remarks. Current studies of the dynamics of steppic vegetation associations in Tunisia are important in order to throw light on the Holocene records and to explain certain changes in the vegetation structure such as those observed in fossil pollen spectra. The elements suggested by the referee have been inserted into our discussion on the dynamics of Artemisia during the Holocene, particularly in relation to the replacement of the *Rhanterium suaveolens* steppe by the *Artemisia campestris* steppe in the Jeffara (Chaieb and Zaâfouri, 2000; Genin et al. 2006). However, notwithstanding the complementarity between contemporary botanical studies and palaeoecological data produced by pollen analysis, as for example in the case of A. campestris and Rhanterium suaveolens, we feel that a significant difference exists in approaching vegetation dynamics within a temporal perspective at the scale of the Holocene through pollen analyses. Thus, for Artemisia, the pollen data from Sebkha Boujmel indicate a relatively early and progressive development, closely linked to anthropic activity, even though other factors could also have played a role in this development. These data do not support the claim for an exclusively contemporary and very recent development. On the contrary, it is important to place the recent development of Artemisia within a long-term dynamic which is also apparent in other pollen diagrams in Tunisia (Brun, 1983; Brun and Rouvillois-Brigol, 1985).
- Manuscript change: (p. 15, l. 23-26): added sentence: "In fact, studies of the dynamics of current steppic vegetation indicate that *Artemisia campestris* is a pioneer species in its occupation of *Rhanterium suaveolens* sandy steppes which have been degraded by vegetation clearance and over-grazing on the Jeffara Plain (Chaieb and Zaâfouri, 2000; Genin et al., 2006).

- Added reference: (p.20, I.25-28): Chaieb, M., and Zaâfouri, M. S.: L'élevage extensif, facteur écologique primordial de la transformation physionomique du cortège floristique en milieu steppique tunisien, in: Rupture : nouveaux enjeux, nouvelles fonctions, nouvelle image de l'élevage sur parcours, edited by: Bourbouze, A., and Qarro, M., CIHEAM, Montpellier, 217-222, 2000.
- Salvadora persica is a species of the Middle East and the Persian Gulf, and has never existed in North Africa.
- Author's response: Salvadora Persica is mentioned with reference to the work carried out by Giraudi and colleagues (2013) who report the occurrence of pollen of this species in the nearby Libyan Jeffara to the south. This species is currently reported from many Saharan mountains such as Hoggar and Tassili (e.g. Ozenda, 2004, p.366). As regards the Holocene palaeo-botanical records, besides the data from the Libyan Jeffara, both pollen (Mercuri, 2008) and charcoal (Neumann and Uebel, 2001) of Salvadora persica are reported from Holocene archaeological sites in the Libyan Sahara.
- Several scientific plant species names are written with errors. example, Haloxylon scoparium not Holoxylon scoparium in the legend to Figure 1. The authors employ often old scientific nomenclature. I invite them to review the names of species according to the new nomenclature, proposed by Le Floc'h, Boulos & Vela (2010). Finally, authors should consider these remarks on the current flora to claim the publication of this work.
- Author's response: All botanical species names in the text and figure captions have been checked for typing mistakes and have been duly corrected in accordance with Le Floc'h et al., 2010. The latter reference has been added to the paper and inserted in the text (P.3, L.25). However, we still refer to *Artemisia herba-alba* (p.15,I.22) in order to make it easier for readers to follow this work in respect to previous published data and studies, and also because the evidence for *Artemisia saharae* is not yet confirmed with certitude in Tunisia (Le Floc'h et al. 2010).
- Manuscript change:
  - o P3, I. 25: added sentence: "(botanical nomenclature is following Le Floc'h et al., 2010)"
  - Correction and check for mistakes for all botanical names:
    - Artemesia corrected to Artemisia (p.4, 1.7; p.15, 1.15)
      - Olea europaea spp. europaea var. sylvestris corrected to Olea europaea subsp. europaea var. sylvestris (p. 8, 1. 26, 1. 31; p.9, 1. 4/5)
      - Acacia raddiana corrected to Acacia tortilis subsp. raddiana (p. 12, 1. 24)
      - Olea europea Corrected to Olea europaea subsp. europaea var. europaea (p. 14, 1. 17)
      - Artemisia herba alba corrected to Artemisia herba-alba (p. 15, 1. 22)
      - Caption figure 1, *Holoxylon* corrected to *Haloxylon*
- Added reference: (p.25, I.30-31): Le Floc'h, E., Boulos, L., and Vela, E.: Catalogue synonymique commenté de la Flore de Tunisie, Ministère de l'Environnement et du Développement durable Banque Nationale de Gènes, Montpellier-Tunis, 500 pp., 2010.

**Anonymous Referee #2**

- Radiocarbon dating: The authors state that the organic matter they dated is "of mixed origin, composed of marine planktonic/algal material and continental woody material" (p.5, l.3-4). In this case, one may wonder if correction for marine reservoir effect is appropriate it may lead to an overcorrection of the dates? It would have been better, of course, to be able to date terrestrial macrofossils instead of bulk organic matter but these are notoriously missing in such environments. However, for further studies, the authors may consider the possibility of picking small particles of charcoal (200-500  $\mu$ m), which may be present in the sediment and may provide more reliable ages. This, and the general aspect of the age-depth model (Fig. 2) indicate that indeed time constrain may not be that reliable below ca. 4000 BP, and this renders it difficult to compare short-term events with other records. Although the authors take great care to emphasis this in the discussion, it may be better to acknowledge this issue by introducing more caution (e.g., by the use of ca.) when summarising the results (e.g. in the abstract).
- Author's response: Thank you for this remark and the suggestion about dating charcoal particles for future research. The referee's advice has been taken on board by using "ca." for all dates within the abstract and the paper text.

As the referee rightly points out, terrestrial organic material suitable for dating is lacking in such sediments. The presence of benthic foraminifera, the mixed origin of the dated materiel, and the absence of data regarding the relative percentages of planktonic/algal material and continental woody material within each sample, obliged us to correct the dates for marine reservoir effect. In all, we think that we have an acceptable chronological framework as maximum age error ranges from 1 to 2 centuries considering the error bars with or without corrections for reservoir effect. Efforts have also been made to obtain a maximum age control within the sequence by cross-checking 14C dates and tephrochronological data. Unfortunately, sieving and observation under binocular microscope do not allow tephra particles to be identified, even though such material has been reported from the nearby Sebkha Mhabeul (Marqueur et al.2008).

- Manuscript change: ca. is added for all centennial scale dates within the text and the abstract.
- Comparison to other records: Roman period: there are actually some indications that the climate may have been colder during the Roman period in the Middle East, so the term "Roman Warm Period" may not be appropriate for the whole region. . . (e.g. Issar 2003 - Climate Changes during the Holocene and their Impact on Hydrological Systems, INTERNATIONAL HYDROLOGY SERIES, Cambridge university press).
- Author's response: We thank the referee for this remark highlighting the problems arising from the use of northern European terminology when dealing with climate change in other regions, especially when this terminology includes climatic indications such as "warm". The "Roman Warm Period" does not have a global signal as an RCC with important regional differences and peculiarities (e.g. Martín-Puertas et al. 2008). Therefore, we use "Roman Warm Period" as a generic term for this climate event and refer to it within the text as a climate episode "commonly known as the Roman Warm Period (RWP)". Based on our data, we do not yet have information as to whether this period was warm or cold and the main element that we discuss is the relative wetness of this period based on our data compared to other records from the southern Mediterranean. Therefore, we have decided to use the term "Roman Humid Period-RHP", as done, for example, by Martín-Puertas et al. 2009 and Jiménez-Moreno et al. 2013 for southern Spain. "Roman Humid Period" seems more appropriate and consistent with our data and other

southern Mediterranean records discussed in the text, such as the Middle East where a colder and more humid climate prevailed during the Roman period (Issar, 2003).

- Manuscript change:
  - **P. 14, I. 30:** "This episode, commonly known as the Roman Warm Period (RWP), is also recorded at other sites." Is replaced by "This episode, the Roman Humid Period (RHP), is also recorded at other sites.
  - **P. 14-15,** RWP is replaced by RHP

**Editor comment:**

- Please add BP to all your dates: ca. 3 ka BP
- Manuscript change: BP is added to all dates within the text

**Other changes to the manuscript:**

- P.5, I.29: (Anabasis, Cornulaca, Haloxylon, Traganum) corrected to (Anabasis, Cornulaca, Haloxylon and Traganum)
- P.15, l. 8 : (850 1459 AD) corrected to (850 1450 AD)
- P. 16, l. 7: Reference to Sadori et al., 2013 is replaced by Sadori et al., 2016
- P.29, l. 29-31, reference to:

Sadori, L., Ortu, E., Peyron, O., Zanchetta, G., Vannière, B., Desmet, M., and Magny, M.: The last 7 millennia of vegetation and climate changes at Lago di Pergusa (central Sicily, Italy), Clim. Past, 9, 1969-1984, 10.5194/cp-9-1969-2013, 2013.

**is replaced by:**

Sadori, L., Giraudi, C., Masi, A., Magny, M., Ortu, E., Zanchetta, G., and Izdebski, A.: Climate, environment and society in southern Italy during the last 2000 years. A review of the environmental, historical and archaeological evidence, Quaternary Science Reviews, 136, 173-188, http://dx.doi.org/10.1016/j.quascirev.2015.09.020, 2016.

- Pollen taxa name have been checked for typing mistakes in figures 3, 4 and 5 captions and table 2:
  - Cornulaca-Traganum type corrected to Cornulaca/Traganum-t.
  - Moltkia ciliata corrected to Moltkiopsis ciliata
  - Typha-Sparganium corrected to Typha/Sparganium-t.
  - *Quercus* type-*ilex* corrected to *Quercus ilex*-t
  - Cerealia t. corrected to Cerealia-t.
  - Acacia t. cyanophilla corrected to Acacia cyanophylla-t.,
  - Figure 5: Nirophilous corrected to Nitrophilous

**Environmental changes, climate and anthropogenic impact in southern-eastern Tunisia during the last 8 kyr**

Sahbi Jaouadi1, Vincent Lebreton1, Viviane Bout-Roumazeilles2, Giuseppe Siani3, Rached Lakhdar4, Ridha Boussoffara5, Laurent Dezileau6, Nejib Kallel7, Beya Mannai-Tayech8, Nathalie Combourieu-Nebout1

[revised manuscript text omitted]